# Outer membrane lipoprotein NlpI scaffolds peptidoglycan hydrolases within multi-enzyme complexes in *Escherichia coli*

Manuel Banzhaf[1,†,‡], Hamish CL Yau[2,†,§], Jolanda Verheul[3,†], Adam Lodge[2,¶], George Kritikos[1], André Mateus[1] (iD), Baptiste Cordier[4] (iD), Ann Kristin Hov[1,††], Frank Stein[1], Morgane Wartel[1], Manuel Pazos[2], Alexandra S Solovyova[5], Eefjan Breukink[6], Sven van Teeffelen[4], Mikhail M Savitski[1,7] (iD), Tanneke den Blaauwen[3,*] (iD), Athanasios Typas[1,7,**] (iD) & Waldemar Vollmer[2,***] (iD)

## Abstract

The peptidoglycan (PG) sacculus provides bacteria with the mechanical strength to maintain cell shape and resist osmotic stress. Enlargement of the mesh-like sacculus requires the combined activity of peptidoglycan synthases and hydrolases. In *Escherichia coli*, the activity of two PG synthases is driven by lipoproteins anchored in the outer membrane (OM). However, the regulation of PG hydrolases is less well understood, with only regulators for PG amidases having been described. Here, we identify the OM lipoprotein NlpI as a general adaptor protein for PG hydrolases. NlpI binds to different classes of hydrolases and can specifically form complexes with various PG endopeptidases. In addition, NlpI seems to contribute both to PG elongation and division biosynthetic complexes based on its localization and genetic interactions. Consistent with such a role, we reconstitute PG multi-enzyme complexes containing NlpI, the PG synthesis regulator LpoA, its cognate bifunctional synthase, PBP1A, and different endopeptidases. Our results indicate that peptidoglycan regulators and adaptors are part of PG biosynthetic multi-enzyme complexes, regulating and potentially coordinating the spatiotemporal action of PG synthases and hydrolases.

**Keywords** bacterial cell envelope; endopeptidase; outer membrane lipoprotein; penicillin-binding protein; peptidoglycan
**Subject Category** Microbiology, Virology & Host Pathogen Interaction

**The EMBO Journal (2020) 39: e102246**

## Introduction

Peptidoglycan (PG) provides bacteria with the mechanical strength to maintain cell shape and resist osmotic stresses. The PG layer or sacculus is a mesh-like structure composed of glycan chains connected by peptides and surrounds the cytoplasmic membrane (CM; Vollmer *et al*, 2008a; Silhavy *et al*, 2010). Given the internal turgor of the cells, PG layer growth requires the coordinated action of synthases and hydrolases to enlarge the sacculus without rupture. This important task is executed by large protein complexes, the elongasome and the divisome, which recruit PG enzymes together with regulators, cytoskeletal, morphogenesis and other structural proteins (Typas *et al*, 2012; Typas & Sourjik, 2015; den Blaauwen *et al*, 2017). It has been previously hypothesized that the formation of these complexes enables the cell to coordinate and regulate the activities of various synthetic and hydrolytic PG enzymes in a spatiotemporal manner (Höltje, 1993). Within these complexes, the key bifunctional penicillin-binding protein (PBP) PG synthases are activated by cognate outer membrane (OM)-anchored lipoproteins (Paradis-Bleau *et al*, 2010; Typas *et al*, 2010, 2012; Dorr *et al*, 2014;

1  European Molecular Biology Laboratory, Genome Biology Unit, Heidelberg, Germany
2  Centre for Bacterial Cell Biology, Biosciences Institute, Newcastle University, Newcastle Upon Tyne, UK
3  Bacterial Cell Biology & Physiology, Swammerdam Institute for Life Sciences, Faculty of Science, University of Amsterdam, Amsterdam, The Netherlands
4  Microbial Morphogenesis and Growth Lab, Institut Pasteur, Paris, France
5  Newcastle University Protein and Proteome Analysis, Newcastle Upon Tyne, UK
6  Membrane Biochemistry and Biophysics, Department of Chemistry, Faculty of Science, Utrecht University, Utrecht, The Netherlands
7  European Molecular Biology Laboratory, Structural & Computational Unit, Heidelberg, Germany
   *Corresponding author. Tel: +31 20 525 3852; Email: t.denblaauwen@uva.nl
   **Corresponding author. Tel: +49 6221 3878156; Email: typas@embl.de
   ***Corresponding author. Tel: +44 191 208 3216; Email: w.vollmer@ncl.ac.uk
   †These authors contributed equally to this work
   ‡Present address: Institute of Microbiology & Infection and School of Biosciences, University of Birmingham, Edgbaston, Birmingham, UK
   §Present address: Faculty of Science, Agriculture and Engineering, Newcastle University, Newcastle Upon Tyne, UK
   ¶Present address: Iksuda Therapeutics, The Biosphere, Newcastle Upon Tyne, UK
   ††Present address: École polytechnique fédérale de Lausanne SV IBI-SV UPDALPE, AAB 013, Lausanne, Switzerland

Egan *et al*, 2014, 2018; Greene *et al*, 2018; Moré *et al*, 2019) and coordinate their action with another, cell constriction-related protein complex (Gray *et al*, 2015). However, with the exception of the amidases (Uehara *et al*, 2010; Yang *et al*, 2012; Peters *et al*, 2013; Tsang *et al*, 2017), it is less clear how Gram-negative bacteria control the activities of their repertoire of hydrolases, i.e. the endopeptidases (EPases), carboxypeptidases (CPases) and lytic transglycosylases.

NlpI is an OM-anchored lipoprotein predicted to be involved in cell division and responsible for targeting the PG EPase MepS for proteolytic degradation (Ohara *et al*, 1999; Singh *et al*, 2015). Deletion of *nlpI* causes cell filamentation at elevated temperature (42°C) or low osmolarity, whilst overexpressing NlpI results in the formation of prolate spheroids (Ohara *et al*, 1999). Deletion of *nlpI* has further implications on the stability of the OM as it increases membrane vesicle formation, in a manner that depends on the activity of two EPases; PBP4 in stationary phase and MepS in exponential phase. This vesicle formation phenotype is suppressed by a deletion of *mepS* (Schwechheimer *et al*, 2015). Many of its pleiotropic effects may be due to the ability of NlpI to target the EPase MepS for proteolytic degradation by forming a complex with the tail-specific protease Prc (Su *et al*, 2017). NlpI and MepS both interact with Prc, but whilst MepS is digested, only 12 C-terminal amino acids of NlpI are removed (Singh *et al*, 2015). In the absence of NlpI, the half-life of MepS increases from ~2 min to ~45 min. Further, in the Δ*nlpI* mutant, uncontrolled levels of MepS have been shown to impair cell growth on low osmolarity medium and lead to the formation of long filaments (Singh *et al*, 2012, 2015).

NlpI forms a homodimer (Wilson *et al*, 2005) with the 33 kDa monomers having their OM-binding N-termini in close proximity. Each monomer consists of 14 α-helices forming 4 canonical but distinct tetratricopeptide helix-turn-helix repeats (TPR) and 2 non-TPR helix motifs. TPR are found in many protein-interacting modules (Zeytuni & Zarivach, 2012). A putative binding cleft is formed from the curvature of the helices on each monomer, which would be available for protein–protein interactions (Das *et al*, 1998; Wilson *et al*, 2005). It is hence possible that NlpI acts as a scaffold for the formation of protein complexes. In this study, we provide evidence that in addition to targeting MepS for degradation, NlpI scaffolds hydrolases within PG multi-enzyme complexes in *E. coli*.

# Results

### Deletion of NlpI alters abundance and thermostability of envelope biogenesis proteins

Deletion of *nlpI* causes several pleiotropic phenotypes and morphological changes. To link the observed phenotypes to changes in protein abundance and activity, we compared an *nlpI* knockout strain (Δ*nlpI*) to wild-type *E. coli* using two-dimensional thermal proteome profiling (2D-TPP; Savitski *et al*, 2014; Mateus *et al*, 2018). In TPP, both protein abundance and thermostability can be measured. The latter depends on the intrinsic physical properties of the protein and on external factors that stabilize its fold, such as protein–protein and protein–ligand interactions.

Numerous proteins changed abundance and thermostability in the Δ*nlpI* cells (Tables EV1 and EV2). In agreement with its

periplasmic location and links to envelope integrity (Schwechheimer *et al*, 2015), deletion of *nlpI* resulted in changes in abundance and thermostability of major envelope components, including outer membrane proteins (OMPs), the β-barrel assembly machinery (BAM; Noinaj *et al*, 2017) and the Tol-Pal complex (Egan, 2018; Fig 1A and B). As expected, both MepS abundance and thermostability were dramatically elevated in Δ*nlpI* cells, since in the absence of NlpI, MepS is not targeted for degradation by Prc (Singh *et al*, 2015; Fig 1A and B). We also observed that other PG biogenesis proteins showed mild increases in abundance and these included several PG hydrolases (PBP5, PBP6a, MltA, MltG), LdtB, LdtF and PG synthases (PBP1A, PBP1B; Fig 1A). A number of these also decreased in thermostability, with lytic transglycosylases (MltA, MltC, MltE), the LD-transpeptidase LdtF and the PG synthases and their regulators (PBP1B, LpoA, LpoB) showing the strongest effects (Fig 1B). In contrast, all amidases (AmiA, AmiB and AmiC) decreased in abundance (Fig 1A). Moreover, the amidase regulator NlpD (which binds to AmiC and controls its activity; Uehara *et al*, 2010) and the YraP protein, which was recently implicated in the activation of NlpD, were strongly destabilized (Fig 1B; Tsang *et al*, 2017).

To ensure that pleiotropic changes are not due to polar gene expression caused by inactivation of NlpI, we complemented the Δ*nlpI* mutant by expressing endogenous NlpI from an arabinose inducible, medium copy number plasmid (pBAD30). The complemented strain restored cell length and partially cell width to wild-type values (Appendix Figs S5F, and S12A and B). The lack of full complementation of cell widths could be due to our inability to precisely restore the level and regulation of NlpI and, consequently, the level of MepS (Ohara *et al*, 1999), Overall, our results indicate that almost all effects in the Δ*nlpI* mutant are due to cells lacking NlpI.

To test whether the observed changes are due to higher abundance of MepS in the Δ*nlpI* mutant, we repeated the 2D-TPP with an Δ*nlpI*Δ*mepS* mutant (Appendix Fig S1A and B). Several of the changes observed in the Δ*nlpI* cells remained in the Δ*nlpI*Δ*mepS* background (Appendix Fig S1A and B), including the destabilization of many cell wall enzymes and regulators. We also directly compared the 2D-TPP profiles of Δ*nlpI* and Δ*nlpI*Δ*mepS* mutants (Appendix Fig S1C and D), with the major difference between both proteomes being that some OMPs were more stable in Δ*nlpI* cells. Importantly, the stability changes occurring for PG enzymes were not observed in this comparison, indicating that they occur independently of MepS levels. Altogether, these results provide the first evidence that NlpI affects PG biogenesis beyond the known interaction with the EPase MepS.

### NlpI pulls down several classes of PG hydrolases and multiple divisome proteins

The decrease in thermostability of several PG biogenesis proteins in Δ*nlpI* cells raised the possibility that NlpI may interact with these proteins. To investigate this further, we applied detergent-solubilized *E. coli* membrane proteins to immobilized NlpI to identify potential interaction partners. Affinity chromatography was performed both in low salt binding conditions (50 mM) to pull down larger PG multi-enzyme complexes, and in high salt binding conditions (400 mM) to

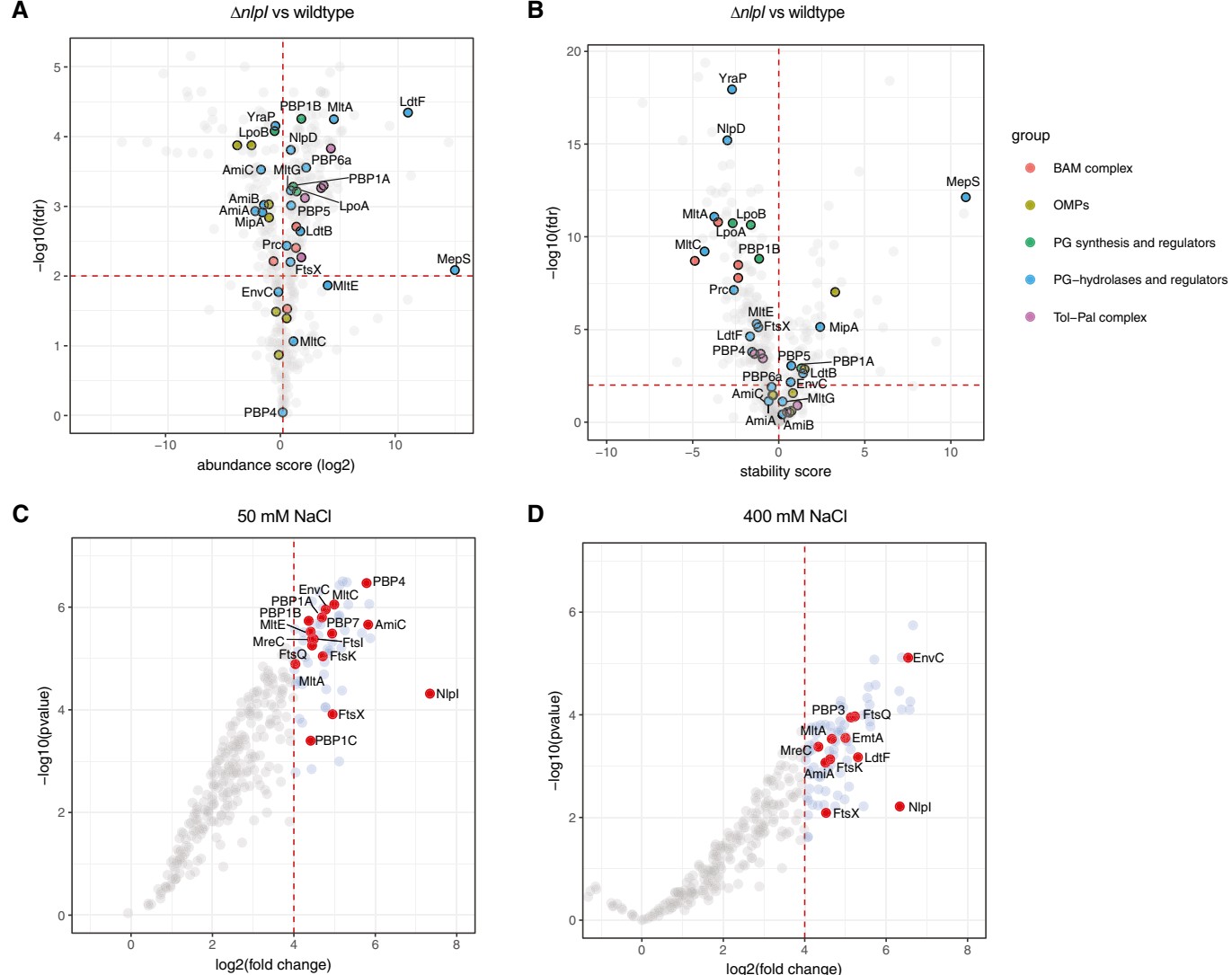

**Figure 1.** *In vivo* and *in vitro* proteomics-based assays link NlpI to several classes of PG hydrolases.

A, B  Wild-type and Δ*nlpI* cells were heated at a range of temperatures, and the soluble components were labelled by TMT, combined and quantified by LC-MS, using the published 2D-TPP protocol (Mateus *et al*, 2018). Shown are volcano plots of two replicates depicting changes in protein abundance (A) and thermostability (B). A local FDR (false discovery rate) < 0.01 was set as a threshold for significance. Highlighted proteins: outer membrane proteins (OMPs, light green), β-barrel assembly machinery (BAMs, red), PG synthases/regulators (green), PG hydrolases and regulators (blue) and the Tol-Pal complex (violet). All other proteins were coloured grey and not labelled to increase the plot clarity. Full results can be found in Tables EV1 and EV2.

C, D  Affinity chromatography with immobilized NlpI. Membrane extracts from *E. coli* were incubated in low and high salt binding conditions (50 and 400 mM NaCl, respectively), and then eluted with 1 M NaCl or 2 M NaCl to identify possible interaction partners by label-free LC-MS analysis. The plot shows the $\log_2$ fold enriched proteins when compared to those eluted from a parallel empty column control, versus the $\log_{10}$ P-value, in low (4 replicates) (C) and high (2 replicates) (D) salt. Highlighted points are all interactions with PG enzymes and their regulators, as well as members of the divisome. All other proteins were coloured grey and not labelled to increase the plot clarity; many were non-physiological interactions with abundant cytoplasmic proteins. Full results can be found in Tables EV3 and EV4. GO enrichments can be found in Tables EV9 and EV10.

identify stronger, salt-resistant and possibly direct binding partners. As a control, we used a column containing Tris-coupled sepharose beads and compared elution fractions with label-free mass spectrometry (Tables EV3 and EV4). To investigate relevant NlpI interaction partners, we first performed gene ontology (GO) enrichment analysis and confirmed that proteins pulled down are enriched in several relevant GO terms, such as "cell wall organization" and "peptidoglycan metabolic processes" (Tables EV9 and EV10). Next, we focused on proteins located in the periplasmic space and highlighted known PG biogenesis proteins (Fig 1C and D). For both affinity chromatography experiments, we were unable to detect the known NlpI binding partner MepS in the applied extract, likely due to its low cellular levels in wild-type cells (Fig 3D).

In low salt binding conditions, NlpI retained several envelope biogenesis proteins, such as the PG synthases PBP1A, PBP1B, PBP1C, the divisome proteins EnvC, PBP3, FtsK, FtsQ and FtsX, the lytic transglycosylases MltA and MltC, the amidase AmiC and the EPases PBP4 and PBP7, amongst others (Fig 1C). This shows that NlpI is able to pull-down full or partial PG-synthase complexes. When challenged in high salt binding conditions, many of the aforementioned interactions were lost. However, immobilized NlpI still retained the divisome proteins PBP3, FtsK, FtsQ and FtsX, the amidase AmiA and its regulator EnvC, and the lytic transglycosylases MltA at 400 mM NaCl, suggesting strong, salt-resistant interactions (Fig 1D).

The *in vivo* proteomics of Δ*nlpI* and the subsequent affinity chromatography revealed strong links of NlpI to several classes of PG hydrolases, PG synthases and divisome proteins. To investigate whether NlpI has a broader role in regulating EPases beyond MepS (Singh *et al*, 2015), we next focused on characterizing the interactions of NlpI with EPases and PG synthases in more detail.

### NlpI dimerizes and interacts with several EPases

To confirm the observed interactions between NlpI and EPases, we performed various biochemical assays. A soluble version of NlpI lacking its membrane anchor was used for all these assays. Firstly, we determined that NlpI is predominantly a homodimer using analytical ultracentrifugation (AUC). The experimentally determined sedimentation coefficient was 4.16 S, which is close to the calculated sedimentation coefficient of 4.52 S, based on the crystal structure of the NlpI dimer (1XNF.pdb; Wilson *et al*, 2005) (Appendix Fig S2A). We measured the apparent dissociation constant ($K_D$) for the NlpI dimer as $126 \pm 9$ nM by microscale thermophoresis (MST): titrating a fluorescently labelled NlpI (fl-NlpI) against a serial dilution of unlabelled NlpI (Fig 2A and Appendix Fig S2B). Binding of the unlabelled NlpI to fl-NlpI resulted in changes to the thermophoretic mobility of fl-NlpI, which is expressed as a change in fluorescence and plotted against ligand concentration to derive the binding affinity. The formation of a dimer by NlpI in solution is consistent with previous work (Su *et al*, 2017). We next tested the specificity of a previously reported interaction between NlpI and the EPase MepS, using MST (Singh *et al*, 2015). We found that NlpI and MepS interacted directly, with an apparent $K_D$ of $145 \pm 52$ nM (Fig 2A and B). NlpI also interacted with MepM and PBP4 with similar apparent $K_D$'s of $152 \pm 42$ nM and $177 \pm 49$ nM, respectively (Fig 2A and Appendix Fig S2B). Assaying for an interaction between NlpI and PBP7 by MST revealed a more complex binding curve, which could only be fit assuming a Hill coefficient of ~ 3 (Appendix Fig S2B). This resulted in an apparent $EC_{50}$ value of $422 \pm 25$ nM and suggested an element of positive cooperativity in the NlpI-PBP7 binding.

We also tested the interactions between NlpI and EPases (MepM, MepS, PBP4 and PBP7) by $Ni^{2+}$-NTA pull-down assays and confirmed the interactions found by MST (Appendix Fig S3A). We could not detect an interaction between NlpI and the carboxypeptidase PBP5 or the lytic transglycosylase Slt, suggesting that NlpI does not interact with all hydrolases in general (Appendix Fig S3A). Using a combination of MST and $Ni^{2+}$-NTA pull-down assays, we also tested for interactions between the EPases. Of the four EPases, which we studied and all possible combinations tested, the only

interactions we found were between MepS-MepM and PBP4-PBP7 (Appendix Figs S2C and S3B).

### NlpI scaffolds trimeric complexes between different EPases

Since NlpI bound multiple EPases, we tested whether NlpI could also form trimeric complexes with them. As a starting point, we tested whether NlpI could scaffold MepS and PBP4 in a fixed concentration MST assay. In the presence of 3 μM NlpI, the normalized fluorescence (FNorm) of fl-MepS increased, confirming the interaction between NlpI and MepS (Fig 2C). In contrast, fl-MepS did not interact with PBP4, even when that was used in excess (30 μM; Fig 2C). Interestingly, fl-MepS was able to bind to a saturated NlpI-PBP4 complex indicating the formation of a trimeric complex between NlpI, PBP4 and MepS (Fig 2C). NlpI pre-incubated with excess BSA did not give the same increase in fl-MepS signal, indicating that the FNorm increase was specific to the binding of NlpI-PBP4 (Fig 2C). We also tested whether NlpI was able to scaffold MepS and PBP7. Fl-MepS could bind pre-incubated NlpI-PBP7 complexes indicating that NlpI can scaffold both EPases and likely has different binding sites for MepS and PBP7 (Fig 2D). Using a three-component $Ni^{2+}$-NTA pull-down assay, we were also able to resolve an NlpI-mediated complex containing PBP7 and MepS (Appendix Fig S3C). The trimeric complexes were not due to direct interactions between the EPases (Appendix Fig S2C and S3B), but rather due to NlpI scaffolding both EPases simultaneously. Thus, NlpI can scaffold at least two different trimeric EPase complexes, with MepS-PBP4 and MepS-PBP7.

### NlpI affects EPase activity of MepM and MepS *in vitro*

Although NlpI interacted with and complexed several EPases, the cellular role of such complexes remained unclear. Hence, we investigated whether NlpI increased or decreased the activity of these EPases using *in vitro* PG digestion assays with purified sacculi or pre-digested muropeptides. EPases cleave the peptide bond between neighbouring peptides, resulting in a decrease in TetraTetra (bis-disaccharide tetrapeptide) muropeptides. Therefore, we quantified the remaining cross-linked PG substrate following incubation with the respective EPase and used the decrease in TetraTetra as an indication of EPase activity (Fig 3A and Appendix Fig S4A). Our results show that NlpI reduced the activity of MepM, which was more active by itself against sacculi. In contrast, MepS was inactive against sacculi and pre-digested muropeptides, but the addition of NlpI slightly activated MepS against muropeptides (Fig 3A; see also methods). We did not observe significant differences in the activity of PBP4 or PBP7 in the presence of NlpI (Fig 3A). These results suggest that NlpI is able to modulate the activity *in vitro* of certain (e.g. MepM and slightly MepS), but not all, EPases.

### NlpI genetically interacts with EPases and its absence alters cell morphology

To address whether NlpI-EPase complexes are relevant for fitness in *E. coli,* we deleted *nlpI* in combination with different EPases and compared the fitness of the double mutants with that of the parental single mutants (Fig 3B). Only *nlpI* and *mepS* exhibited a strong positive genetic interaction with the double-mutant Δ*nlpI*Δ*mepS*

**A**

| Interaction with NlpI | Apparent $K_D$ / $EC_{50}$ (nM) |
|---|---|
| NlpI | 126 ± 9 |
| PBP4 | 177 ± 49 |
| PBP7 | 422 ± 25 |
| MepM | 152 ± 42 |
| MepS | 145 ± 52 |

**B**

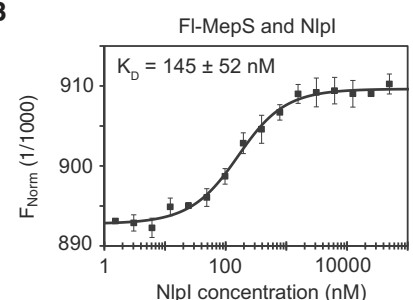

**C**

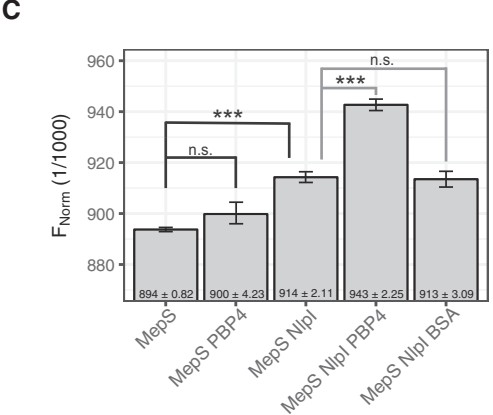

**D**

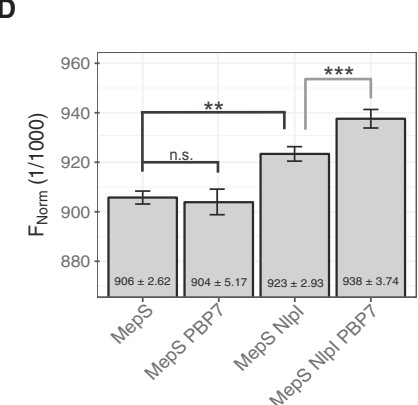

**Figure 2. NlpI interacts with several EPases and is able to form trimeric complexes with them.**

A Dissociation constants for interactions between NlpI with MepM, MepS, PBP4, PBP7 as determined by microscale thermophoresis (MST). The values are mean ± SD of three independent experiments. The corresponding MST binding curves are shown in Appendix Fig S2B.

B MepS-NlpI interaction by MST as an example plot for Fig 2A. The same plot is also shown in Appendix Fig S2B. MST curve plotted is the mean data ± SD of three independent experiments. Fl, fluorescently labelled; FNorm, normalized fluorescence.

C, D NlpI has different binding sites for MepS and PBP4, and MepS and PBP7 as shown by the ability of labelled MepS to bind pre-formed NlpI-PBP4 (C) and NlpI-PBP7 (D) complexes by a fixed concentration MST assay. Values are mean ± SD of 3–6 independent experiments. To calculate significance, the data were fit using a linear model. Calculated means were compared using Tukey's HSD test, resulting in *P*-values corrected for multiple testing. Relevant *P*-values are highlighted directly in the figure (*< 0.05; **< 0.01, ***< 0.001), and all *P*-values can be found in Table EV7.

growing as well as the Δ*mepS* mutant, and better than the Δ*nlpI* mutant. The other mutant pairs exhibited none to very mild genetic interactions based on fitness assays (Fig 3B). To investigate whether more genetic interactions existed but were not visible in fitness assays, we looked for changes at the single-cell level. To do this, all NlpI-EPase single and double mutants were grown exponentially and their morphology was assessed using phase-contrast microscopy (Appendix Fig S3C and S12C). First, we noticed that all tested Δ*nlpI*ΔEPase double mutants were almost as thin as Δ*nlpI* cells (i.e. 5–10% thinner than wild type—hence no genetic interaction), except for the Δ*nlpI*Δ*mepS* mutant, which was even wider than the Δ*mepS* single mutant (Fig 3C). This strong genetic interaction is in line with the fitness data (Fig 3B) and further points to the phenotypes in the *nlpI* mutant being beyond mis-regulated MepS. Some more subtle genetic interactions between *nlpI* and EPases were also apparent. Deleting *nlpI* in Δ*mepM* mutants produced a subpopulation of filamentous cells (Appendix Fig S12C). We also noted that Δ*pbpG* mutants were shorter and fatter than wild type, but double mutants with Δ*nlpI* exhibited only the expected additive effects (Fig 3C and Appendix Fig S12C).

To further assess how much MepS levels interfere with the Δ*nlpI* phenotype, we constructed an arabinose-inducible MepS plasmid (pBAD30). We first confirmed that MepS is overexpressed and does not cause strong fitness defects (Fig 3D and Appendix Fig S13A). Next, we investigated whether MepS expression contributes to morphological changes. Overexpression of MepS increased cell length and slightly reduced cell width (Fig 3E and F, Appendix Fig S13B and C), although not to the level of Δ*nlpI* mutants (Fig 3C). Hence, Δ*nlpI* mutants and MepS overexpression strains share the dramatic increase in MepS levels (Figs 1A and 3D) but the cell morphology changes only to a certain extent. This further supports that the Δ*nlpI* mutant phenotypes can be partially (but not fully) explained by elevated MepS levels.

To further investigate whether the *nlpI* phenotypes go beyond elevated MepS levels, we expanded the fitness genetic interaction assays in selected growth conditions (Fig 3G). In low osmolality medium (LB medium without salt), the Δ*nlpI* mutant was very sick compared to wild-type cells (fitness ratio 0.31), likely due to increased turgor pressure. This could be rescued by deleting *mepS*, up to the fitness levels of the Δ*mepS* mutant (fitness ratio 0.77). In

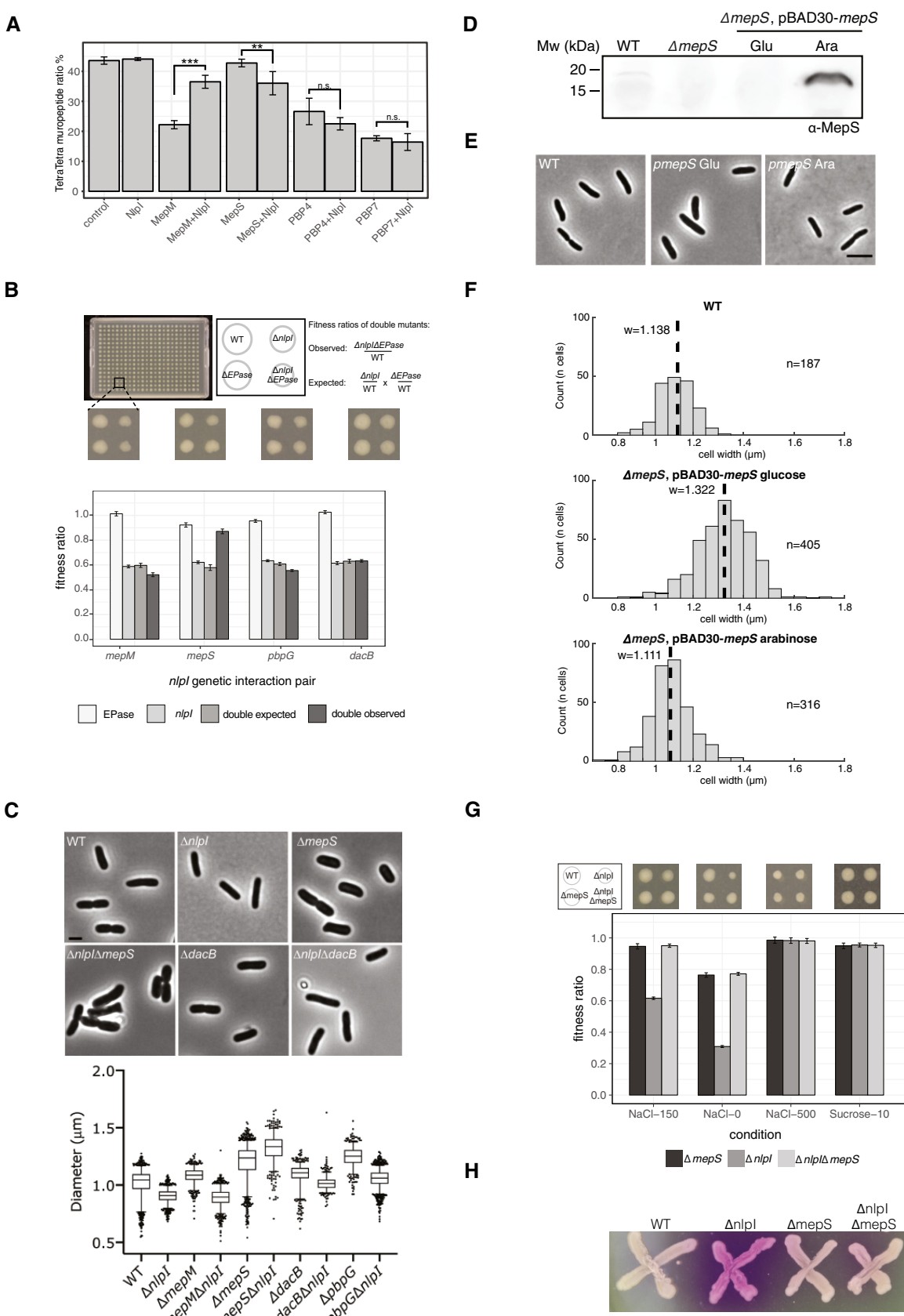

Figure 3.

**Figure 3.  NlpI genetically interacts with MepS and affects the enzyme activity of MepS and MepM.**

A   HPLC-based PG digestion assay representing EPase activity. The graph shows the relative percentage of the muropeptide TetraTetra present at the end of the incubation period for each protein as described in Materials and Methods. MepM and PBP4 were incubated with sacculi, whilst MepS and PBP7 were incubated with soluble muropeptides, both from *E. coli* MC1061, respectively. Values are mean ± SD of three independent experiments. Representative chromatograms are shown in Appendix Fig S4. To calculate significance, the data were fit using a linear model. Calculated means were compared using Tukey's HSD test, resulting in *P*-values corrected for multiple testing. Relevant *P*-values are highlighted directly in the figure (*< 0.05; **< 0.01, ***< 0.001), and all *P*-values can be found in Table EV8.

B   Genetic interactions of *nlpI* with EPase genes. Strains were arrayed using a Rotor HDA replicator on Lennox LB agar plates and incubated for 12 h at 37°C. Each plate contained 384 colonies, 96 from the wild type, single mutants and double mutants. An example of a 384-well plate is shown. Double mutants were made twice, swapping the resistance markers to the two single mutants. Colony integral opacity was quantified as a fitness readout, using the image analysis software Iris (Kritikos *et al*, 2017). Bar plots show the averaged values of 2 biological experiments, each having 96 technical replicates ($i$ = 2, $n$ = 192). The error bars represent the 95% confidence interval. Full results can be found in Table EV5.

C   *nlpI* deletion changes the morphology of EPase-mutant strains. The graph shows the cell width of single- and double-deletion strains (800 < $n$ < 2,000 cells). The box has a medium between 25 and 75%. The whiskers with the upper and lower vertical line indicating the 95 and 5%. The dots are individual points outside the 5 and 95% range. Above the graph are representative images of cells lacking MepS or PBP4 in combination with a deletion of NlpI. The same images for control and NlpI mutant strains have been reused in Fig 4C. The scale bar equals 2 µm. Gene encoding protein legend: *nlpI* encodes NlpI, *dacB* encodes PBP4, *pbpG* encodes PBP7, *mepM* encodes MepM, *meps* encodes MepS. Cell length of mutants is displayed in Appendix Fig S12c.

D   Inducible *mepS* expression system (pBAD30) strongly overproduces MepS. Strains were grown in LB at 30°C, and cells were collected at OD600≈0.4. The level of MepS contained in the membrane fraction was detected using purified anti-MepS antibody.

E   Visualization of the effect of MepS absence or its overexpression on cell width by phase-contrast microscopy. Cultures were grown in LB at 30°C, and aliquots of culture were taken at OD600≈0.1. The scale bar equals 5 µm.

F   MepS level modulates cell width. The graph shows the distribution of mean cell width for each cell, with n corresponding to the number of cells measured for each strain and the median width for the population being indicated by a dotted line and referred to as w.

G   Relative fitness of Δ*nlpI*, Δ*mepS* and Δ*nlpI*Δ*mepS* mutants. Strains were arrayed using a Rotor HDA replicator on Lennox LB agar plates supplemented 10% sucrose, or LB agar plates containing 0 mM or 500 mM NaCl. Plates were incubated for 12 h at 37°C. Each plate contained 384 colonies, 96 from the wild type, single mutants and double mutants. Fitness ratios, bar plots and error bars were calculated/made as in (B). Full results can be found in Table EV5.

H   Cells of wild type (WT), Δ*nlpI*, Δ*mepS* and Δ*nlpI*Δ*mepS* containing multicopy plasmids with *lacZ* were grown onto CPRG indicator agar to assay envelope integrity. CPRG (yellow) cannot penetrate intact Gram-negative envelopes. Its conversion by intracellular β-galactosidase to CPR (red) indicates loss of envelope integrity.

contrast, in high osmolality medium (LB medium with 500 mM salt or LB with 10% sucrose) the Δ*nlpI* mutants' fitness was restored to wild-type levels (Fig 3G), and knocking out *mepS* did not cause any further effects. Next, we tested if fitness phenotypes correlate to defects in the envelope integrity of the tested mutants by using a red-β-D-galactopyranoside (CPRG) envelope integrity assay (Paradis-Bleau *et al*, 2014). CPRG is a β-galactosidase substrate that fails to penetrate wild-type cells, therefore being inaccessible to cytoplasmic β-galactosidase, which can hydrolyse CPRG and produce a red colour (CPR). The production of CPR can be used as a readout for envelope permeability and/or cell lysis. Knocking out *mepS* restored the envelope integrity defects seen in the Δ*nlpI* mutant (Fig 3H). Thus, in all our fitness assays the increased MepS levels are the cause for the envelope integrity effects observed in the Δ*nlpI* mutant.

In summary, our results provide evidence that cellular MepS levels need to be tightly regulated by NlpI (and Prc), as imbalance causes morphological changes, reduced envelope integrity and fitness. However, although the fitness and envelope integrity defects of the Δ*nlpI* mutant can be fully attributed to elevated MepS levels (at least in assays and conditions we tested), the cell morphology phenotypes (Fig 3C and F) and the global changes in protein abundance and stability (Appendix Fig S1) cannot. Both point to MepS-independent effects in the *nlpI* mutant. In agreement with this, *nlpI* seems to also genetically interact with other EPases (*mepM*) at the least at a morphological level (Appendix Fig S12C). Thus, we conclude that NlpI has additional effects on controlling cell shape beyond the described proteolytic regulation of MepS (Singh *et al*, 2015).

## NlpI localizes along the entire cell envelope

To understand further the physiological role of NlpI, we investigated its cellular localization using specific antibodies. NlpI localized in

the entire envelope and not specifically at midcell (Fig 4A and Appendix Fig S5), in contrast to what its interaction with some divisome proteins suggested (Fig 1D). In addition, its concentration remained constant during the cell cycle (Appendix Fig S5A). To control for possible epitope occlusion by interaction partners of NlpI, we localized a functional C-terminal fusion of NlpI with an HA-tag expressed from a plasmid in the Δ*nlp*I strain. The NlpI-HA localization pattern was identical to that of NlpI (Appendix Fig S5D). We noticed that the localization pattern of NlpI was reminiscent of the PBP synthases PBP1A and PBP2 (den Blaauwen *et al*, 2003; Banzhaf *et al*, 2012). Together with the links of NlpI to PG synthases observed in TPP and pull-downs (Fig 1), this made us wonder whether NlpI-EPase complexes can be part of PG machineries.

## NlpI associates with PG machineries

To probe for genetic interactions with the PG synthetic machineries, we deleted *nlpI* in combination with different PBPs and their regulators (Lpos) and compared the fitness of the double mutants with that of the parental single mutants (Fig 4B). We noticed an almost synthetic lethality with Δ*mrcB* (encodes PBP1B) and Δ*lpoB* (encodes LpoB), fitness ratio of −0.62 and −0.49, respectively. Δ*mrcA* (encodes PBP1A) and Δ*lpoA* (encodes LpoA) also exhibited strong negative interactions with Δ*nlpI*, fitness ratio of −0.30 and −0.21, respectively (Fig 4B). To analyse whether these strong negative genetic interactions were also reflected in the morphology of the cells, all single and double mutants were grown exponentially and imaged by phase-contrast microscopy. Combining Δ*nlpI* with Δ*mrcB* or Δ*lpoB* led to abnormal cell morphologies, with cells being 30% wider and up to 80% longer (Fig 4C and Appendix Fig S12D). This suggests that the NlpI-EPase complexes might be important for facilitating the formation of the PBP1A-mediated PG machinery. This

**A**

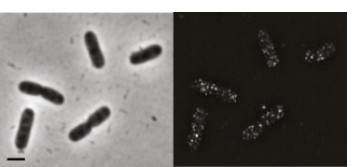

**B**

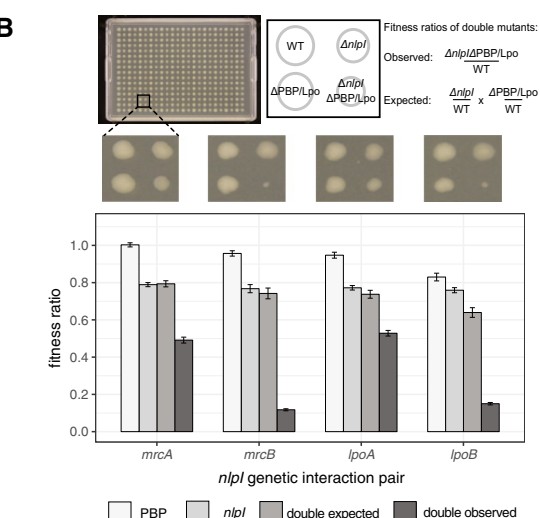

**C**

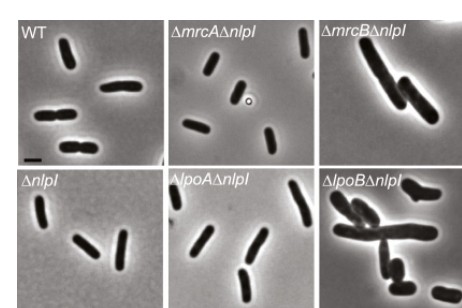

**D**

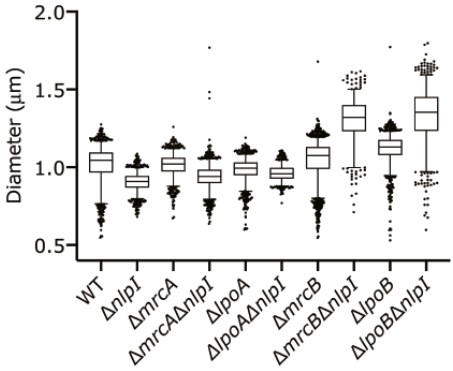

| Ligand | $K_D$ / $EC_{50}$ of interactions (nM) with | |
|---|---|---|
| | PBP1A [1] | LpoA [2] |
| NlpI | No interaction | No interaction |
| MepM | No interaction | No interaction |
| MepS | 91 ± 39 | No interaction |
| PBP4 | 106 ± 44 | 315 ± 38 |
| PBP7 | 101 ± 35 | 217 ± 93 |

**Figure 4. NlpI localizes along the entire cell envelope and associates with PG machineries.**

A Phase-contrast image and corresponding fluorescence SIM image of BW25113 cells that have been grown in LB at 37°C and immunolabelled with specific antibodies against NlpI. Scale bar equals 2 μm. See Appendix Fig S5 for further details.

B Genetic interactions of NlpI with PG machineries. Strains were arrayed and assessed as in Fig 3b. An example of a 384 plate is shown. Bar plots show the averaged values of 2 experiments ($i$ = 2, $n$ = 192). Error bars denote the 95% confidence interval of the mean. Full results can be found in Table EV5.

C NlpI deletion exacerbates the morphological defects of the PBP1B/LpoB-mutant strains. The graph shows the cell width of single- and double-deletion strains (800 < $n$ < 2,000 cells). The box has a medium between 25 and 75%. The whiskers with the upper and lower vertical line indicating the 95 and 5%. The dots are individual points outside the 5 and 95% range. Representative images of the strains are shown above the graph. The same images for control and NlpI mutant strains have been reused in Fig 3C. The scale bar equals 2 μm. Gene encoding protein legend: *nlpI* encodes NlpI, *mrcA* encodes PBP1A, *lpoA* encodes LpoA, *mrcB* encodes PBP1B, *lpoB* encodes LpoB. Cell length of mutants is displayed in Appendix Fig S12D.

D Dissociation constants for interactions between PBP1A and LpoA with NlpI, MepM, MepS, PBP4 and PBP7 as determined by MST. The values are mean ± SD of three independent experiments. [1]PBP1A was used as fluorescently labelled protein in all assays. [2]LpoA was used as unlabelled ligand in all combinations, except with MepM and PBP4. Binding curves are shown in Appendix Fig S6.

would be consistent with the changes in thermostability of PBP1A and LpoA in Δ*nlpI* cells (Fig 1B). Thus, we next tested the *in vitro* interactions between NlpI and respective EPases with PBP1A and LpoA. We discovered that PBP1A did not directly interact with NlpI but interacts with low nanomolar range affinities with different EPases, including MepS (apparent $K_D$ = 91 ± 39 nM), PBP4 (106 ± 44 nM) and PBP7 (101 ± 35 nM) (Fig 4D, Appendix Fig S6A and S7). PBP4 (315 ± 38 nM) and PBP7 (217 ± 93 nM) also bound to LpoA at slightly higher nM ranges (Fig 4D, Appendix Fig S6B and S7). These interactions between PG synthases and EPases would allow for PG multi-enzyme complexes to exist as postulated by Höltje (Höltje, 1998).

## NlpI is part of a PG multi-enzyme complex with PBP4 and PBP1A/LpoA

To further understand the interaction between PG hydrolases and synthases, we characterized in detail the interactions between PBP4 with PBP1A/LpoA and NlpI by MST. We used a fixed concentration MST assay to show that fluorescently labelled PBP1A and LpoA are able to bind a pre-formed PBP4-NlpI complex (Fig 5A and B, Appendix Fig S8A). Whilst the binding of PBP4 and PBP4-NlpI to fl-PBP1A resulted in an increase in FNorm values (which was not the case in the presence of NlpI alone), binding of PBP4 and PBP4-NlpI to LpoA consistently resulted in an enhanced initial fluorescence. This indicated that the ligand was binding in close proximity to the probe and was affecting the local environment of the fluorophore and subsequently its fluorescence yield. Since the change in fluorescence was due to ligand binding (Appendix Fig S8B), the raw fluorescence data as opposed to the FNorm values were plotted in this instance (Fig 5B). These consistent increases in fluorescence reflect the binding of PBP4 and PBP4-NlpI to LpoA and suggest that the presence of NlpI does not prevent the interaction of PBP4 with LpoA

(Fig 5B). Following on from the previous identification of a multi-enzyme complex containing the synthase PBP1B, the lytic transglycosylase MltA and the OM scaffold protein MipA (Vollmer *et al*, 1999), this is the only other biochemical evidence, to our knowledge, that PG synthases and PG hydrolases form multi-enzyme complexes with regulatory lipoproteins to possibly coordinate PG synthesis in Gram-negative bacteria.

## Discussion

*Escherichia coli* contains a repertoire of more than 20 periplasmic hydrolases providing specificity to almost every bond present in PG (Vollmer *et al*, 2008b; van Heijenoort, 2011; Singh *et al*, 2012; Yunck *et al*, 2016; Chodisetti & Reddy, 2019). However, with the exception of amidases, it is unclear how these hydrolases are regulated to prevent autolysis (Uehara *et al*, 2009). This study identifies NlpI as a novel scaffolding protein of EPases that might coordinate hydrolases within PG synthesis machineries. NlpI is also able to bind several other hydrolytic enzymes, including some members of the amidase and lytic transglycosylase families. The details of these interactions will be investigated in future work.

### Deletion of *nlpI* impacts envelope biogenesis beyond the proteolytic regulation of MepS levels

NlpI interacts with MepS and targets it for degradation via the protease Prc (Singh *et al*, 2015). Inactivation of *mepS* leads to a 17% increase in cell diameter compared to wild-type cells (Fig 3C), whereas overexpression of *mepS* reduced the cell diameter, although not to the level of Δ*nlpI* mutants (Fig 3C and F). Nevertheless, the observed shape changes provide further evidence that cellular MepS levels impact the cell diameter. On the other hand, inactivating both, *nlpI* and *mepS,* increased the cell diameter up to 30% compared to wild-type cells (Fig 3C). Therefore, Δ*nlpI*Δ*mepS* mutants did not phenocopy Δ*mepS* or Δ*nlpI* mutants in their shape (Fig 3C and Appendix Fig S12C) and this indicates that inactivation of *nlpI* leads to additional morphological effects. This is supported by the observation that the Δ*mepM*Δ*nlpI* cells contain long filaments, a phenotype not seen with either the parental single mutants or the strain overexpressing MepS. In conclusion, the contribution of NlpI to cell morphology goes beyond mis-regulated MepS levels.

Δ*nlpI* mutants are known to increase OM vesicle formation (Schwechheimer *et al*, 2015) and shown here to have reduced fitness (especially in hypoosmotic conditions) and envelope integrity compared to wild-type cells (Fig 3B, G and H). All these effects are due to elevated MepS levels, as they are fully resolved in the Δ*nlpI*Δ*mepS* mutant. Thus, the envelope integrity defects of the Δ*nlpI* mutants are mainly (if not entirely) due to elevated MepS levels.

In summary, we show that the interplay of NlpI-MepS impacts fitness, cell morphology and envelope integrity. However, Δ*nlpI*Δ*mepS* mutants do not phenocopy Δ*nlpI* or Δ*mepS* mutants and in addition differed in many of the global changes in protein abundance and stability compared to the Δ*nlpI* mutant or wild-type cells. In addition, our biochemical evidence (protein–protein interactions and protein activity assays) and genetic interactions suggest that NlpI binds to a number PG hydrolases and synthases and their regulators, affecting PG-related processes. NlpI binds to and inhibits MepM *in vitro*, which is reflected by a positive genetic interaction *in vivo*. NlpI also binds strongly to amidases and their regulators (AmiA, EnvC; Fig 1B and D), lytic transglycosylases (MltA, MltC; Fig 1B and D) and other EPases (PBP4, PBP7; Fig 2, Appendix Fig S2 and S3), some in the context of PG biosynthetic machineries (Figs 4–6). This raises the possibility that NlpI scaffolds, or even regulates, several classes of hydrolases beyond its function towards EPases. To the best of our knowledge, this is the first evidence that NlpI has additional functions in PG synthesis.

### NlpI interacts with several EPases at physiologically relevant concentrations

Immobilized NlpI retained the EPases PBP4 and PBP7, raising the possibility that NlpI interacts with additional EPases along with MepS (Fig 1C). Especially since MepS was not amongst the proteins being pulled down, despite being known to bind to NlpI (Singh *et al*, 2015), we decided to investigate this further. Using MST and pull-down assays, we validated interactions between NlpI and 3 other DD-EPases; MepM, PBP4 and PBP7, all of them with apparent $K_D$ or $EC_{50}$ values in the nanomolar range (Fig 2C and Appendix Fig S2A). We estimated the concentration of these proteins in the periplasm, assuming cell dimensions of $4.77 \times 10^{-6}$ m (length) and $1.084 \times 10^{-6}$ m (diameter), with a periplasmic width of $21 \times 10^{-9}$ m (Beveridge, 1995; Banzhaf *et al*, 2012; Fig 6A). We conclude that the NlpI-EPase interactions identified in the present work are all, in principle, able to occur in the cell (Fig 6A). Furthermore, our data showed that NlpI could also affect the activity of some of these EPases; for example, the activity of MepM against intact sacculi was reduced in the presence of NlpI (Fig 3A). As NlpI facilitates the proteolytic degradation of MepS (Singh *et al*, 2015), NlpI could be generally restricting the role of cell elongation-related EPases (Singh

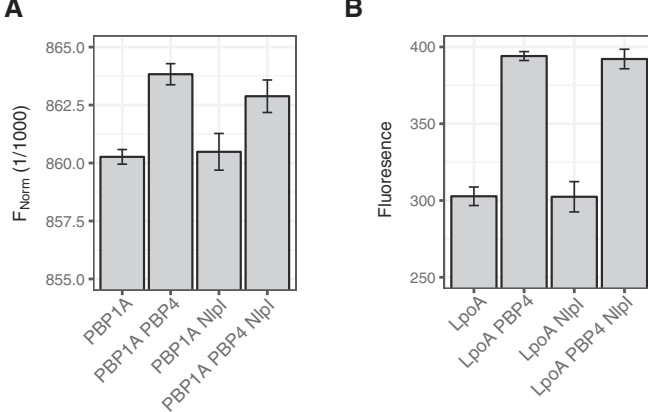

**A** **B**

**Figure 5. PBP4 forms a PG multi-enzyme complex with NlpI and PBP1A/LpoA.**

A, B PBP4 has different interaction sites for PBP1A/LpoA and NlpI as shown by a single concentration MST assay. Plots show the FNorm or fluorescence values of fluorescently labelled PBP1A or LpoA with or without PBP4, NlpI or PBP4-NlpI. Values are mean ± SD of three independent experiments.

**A**

| Protein | No. of molecules per generation[1] | Molarity in periplasm (μM)[2] | Abundance in Δnlpl vs WT (log2) | Stability in Δnlpl vs WT (log2) |
|---|---|---|---|---|
| PG synthesis machineries | | | | |
| PBP1A | 554 | 2.8 | 0.9 | 2.9 |
| PBP1B | 512 | 2.6 | 1.6 | 8.8 |
| LpoA | 513 | 2.6 | 1.2 | -2.7 |
| LpoB | 1490 | 7.4 | -0.7 | -1.6 |
| Nlpl | 389 | 1.9 | N/A | N/A |
| Endopeptidases | | | | |
| PBP4 | 441 | 2.2 | -0.01 | -1.5 |
| PBP7 | 1005 | 5 | NC[4] | NC[4] |
| MepA | 625 | 3.1 | NC[4] | NC[4] |
| MepH | 265 | 1.3 | NC[4] | NC[4] |
| MepM | 341 | 1.7 | NC[4] | NC[4] |
| MepS | 3931 | 19.7 [3] | 15 | 10.9 |

**B**

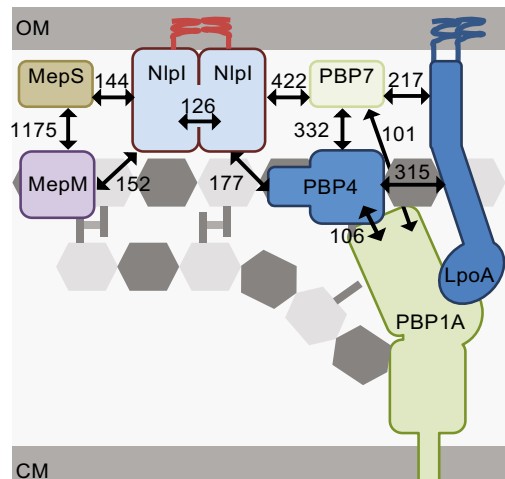

**Figure 6. Proposed model for a role of NlpI in coordinating formation of PG multi-enzyme complexes containing EPases.**

A Estimated number of molecules and molarity of PBP1a/LpoA and EPases. [1]Numbers obtained by ribosomal profiling in rich growth medium (Li et al, 2014). [2]Concentration of monomer. [3]Decreases in the presence of NlpI (Singh et al, 2015). [4]"no change detected". The periplasmic concentrations of proteins were estimated for a cell with periplasmic volume of $3.33 \times 10^{-16}$ l, where 1 molecule corresponds to 5 nM.

B Hypothetical model of NlpI scaffolding endopeptidases during cell elongation. Black arrows indicate interacting proteins with numbers indicating apparent $EC_{50}/K_D$ values. OM, outer membrane; CM, cytoplasmic membrane. MepS–PBP1A interaction is not represented due to illustrative restrictions.

et al, 2012). Consistently, the Δnlpl ΔmepM was more filamentous than its parental single mutants (Appendix Fig S12C).

With regard to activity of EPases, we note that we were unable to observe the DD-EPase activity of MepS, previously reported in Singh et al (2012) (Fig 3A). However, whilst addition of NlpI had no effect on the activity of PBP4 or PBP7, there was a very slight stimulation of MepS activity against isolated muropeptides in the presence of NlpI, following overnight incubation (Fig 3A; see also methods). Overall, these results raise the possibility that NlpI could modulate the activity of specific hydrolases along with its role as a scaffolding protein.

### NlpI scaffolds multi-protein complexes with PG hydrolytic enzymes within the context of PG biosynthesis machineries

NlpI is able to form trimeric complexes with different EPases that lack mutual interactions. Examples of such complexes resolved in the present work are MepS-NlpI-PBP4 and MepS-NlpI-PBP7 (Fig 2C and D). Since NlpI has four helix-turn-helix TPR-like repeats per monomer, it remains to be seen whether the different TPR helixes are specific for different binding partners (Wilson et al, 2005) and/or different type of hydrolytic enzymes. Nevertheless, the ability of NlpI to bind multiple ligands simultaneously is consistent with the idea that TPR domains facilitate the formation of multi-protein complexes (Blatch & Lassle, 1999; Cortajarena & Regan, 2006). In this sense, NlpI is more promiscuous in nature than the previously identified amidase regulators EnvC and NlpD, which have specificity to their cognate amidases (Uehara et al, 2009).

Despite binary interactions between various EPases and PBP1A/ LpoA being able to occur in the absence of NlpI (Fig 4D), we

hypothesize that NlpI could sequester additional or specific sets of EPases and other hydrolytic enzymes, determining the specificity of such synthetic machineries. Accordingly, our finding that PBP4 is able to simultaneously bind PBP1A/LpoA and NlpI supports the idea that NlpI could specifically scaffold hydrolases at active PG synthases (Fig 5A and B). The ability of an OM-anchored NlpI to complex EPases and other hydrolases would not only serve to locally concentrate those enzymes near PG synthesis complexes, but also to maintain the active hydrolases in the space between the PG layer and OM, facilitating cleavage of the mature PG of the sacculus and keeping them at distance to the newly synthesized PG, which emerges between the CM and PG layer and is not subject to turn-over. NlpI molecules are outnumbered by the amount of potential binding partners in the periplasm, so it is unlikely that there is an abundance of free NlpI (Fig 6A). EPase regulation might occur on the level of binding affinity to NlpI and its TPR-like domains. This would see NlpI resembling a "dock" for EPases (and possibly other hydrolases) to make them available for PG synthesis complexes when needed. Such a system would allow for greater flexibility, as NlpI interacts with many hydrolases. Alternatively, the specificity could be encoded on the level of the hydrolases. As demonstrated, EPases interact directly with PG synthases, but those interactions might be specific to particular EPases (and no other hydrolases) and/or might be subject to environmental cues or to competition for the same binding site. Therefore, NlpI could be a more general adaptor of hydrolases, as suggested by its interactions with amidases and lytic transglycosylases (Fig 1B and D), bringing a set of hydrolases to biosynthetic complexes. It is worth noting that loss of nlpI has no significant effects on PG composition (Appendix Fig S11), suggesting the primary role of NlpI is in the coordination of

multi-protein complex formation and not in the regulation of a specific hydrolase. It will require more work to test this hypothesis in the future.

Around 20 years ago, Höltje hypothesized that growth of the PG sacculus requires both synthases and hydrolases working in tandem to enable a safe and coordinated enlargement (Höltje, 1998). However, it has also been suggested that EPases are not necessarily part of multi-protein complexes, as overproduction of three different EPases confers mecillinam resistance (Lai *et al*, 2017). In this work, we provide the first evidence of interactions between PBP1A/LpoA with PBP4 and hypothesize that interactions between NlpI and other EPases could facilitate their delivery to PG synthesis complexes during PG growth. The existence of PG multi-protein complexes is not mutually exclusive with the idea that EPases and/or NlpI-EPase complexes may in part also localize outside of such PG assembly machineries. This work, and the work of others, supports the idea that PG multi-protein complexes are highly dynamic and driven by transient protein–protein interactions (Pazos *et al*, 2017). We note that the respective EPases also have known interactions with other PG processing enzymes, beyond NlpI (Romeis and Höltje, 1994; von Rechenberg *et al*, 1996), and these interactions could also be scaffolded by NlpI or other adaptor proteins, to contribute to the coordination of their activity within complexes. In addition, the existence of such PG multi-protein complexes is in line with the previous isolation of an 1 MDa cell division complex (Trip & Scheffers, 2015).

### NlpI functions together with the PBP1A/LpoA PG machinery

We studied the localization of NlpI to infer whether NlpI scaffolds complexes exclusively for cell elongation or division. The localization pattern of NlpI is spotty and diffusive with no enrichment at the midcell (Fig 4A). NlpI was previously shown to be located in the OM of bacterial cells and is a known lipoprotein (Ohara *et al*, 1999; Teng *et al*, 2010). It is hence also possible that interactions between NlpI and hydrolases concentrate and facilitate cleavage from the outer face of the PG layer. Its disperse localization would enable binding of EPases involved in both division and elongation. NlpI was shown to bind a number of essential divisome proteins at high salt concentrations in our affinity chromatography experiment (Fig 1D). This is consistent with the finding that Δ*nlpI* mutants were initially discovered and classified as a filamentous temperature-sensitive (fts) protein linking it to be conditionally essential for cell division (Ohara *et al*, 1999). The same filamentous phenotype can be observed in low osmolarity (Ohara *et al*, 1999). However, NlpI localization was not enriched at the septum (Fig 4A), suggesting that NlpI might be a transient member of the divisome or interacts with cell division proteins away from midcell in non-dividing cells, and NlpI could also have a main role in PG synthesis during cell elongation. Indeed, the negative genetic interactions of *nlpI* with *mrcA* (PBP1A) and *mrcB* (PBP1B) raise the possibility that NlpI can affect both the elongasome and the divisome. This is in line with both PBP1B/LpoB and PBP1A/LpoA complexes showing changes in thermostability in Δ*nlpI* cells (Fig 1B). However, because the genetic interactions of *nlpI* with *mrcB* (PBP1B) were stronger and led to a near synthetic lethality, we reasoned that NlpI predominantly worked with the PBP1A/LpoA machinery. Cells lacking PBP1B depend on a functional PBP1A/LpoA complex to achieve growth (Yousif *et al*, 1985). An alternative scenario is that cells with

only PBP1A/LpoA are more sensitive to genetic and chemical perturbations in the cell wall than cells with only PBP1B/LpoB, because the latter is more efficient. Although we found no direct interaction between NlpI with PBP1A or LpoA by MST assay (Fig 4D and Appendix Fig S6), a complex of NlpI-PBP4-PBP1A could be formed with PBP4 as the linking protein (Fig 5). The multitude of interactions between PBP1A/LpoA, different EPases and NlpI (Fig 4D and Appendix Fig S6) could enable the formation of different active synthase–hydrolase complexes under a range of growth conditions or availability of particular proteins (Pazos *et al*, 2017).

We note that NlpI in the presence or absence of catalytically inactive MepS(C68A) (MepS*) did not affect the GTase nor TPase activities of PBP1A-LpoA (Appendix Figs S9 and S10, Table EV6), at least in the conditions we tested. This suggests that NlpI's importance to the PBP1A/LpoA system is not as clearly discernible *in vitro* as it is *in vivo* (Fig 4) and this may reflect a more dynamic role as an adaptor protein facilitating the formation of multi-component complexes *in vivo*.

In conclusion, this work provides the first evidence for NlpI as a novel adaptor of EPases (and possibly other classes of PG hydrolases) and we hypothesize that NlpI acts as a scaffolding protein to facilitate the formation of complexes between PG synthases and EPases (Fig 6B).

## Materials and Methods

### Media and growth conditions

Strains used in this work were grown in LB medium (1% tryptone, 1% NaCl, 0.5% yeast extract) at 37°C, unless otherwise stated. Antibiotics were used at the following concentrations (μg/ml): ampicillin (Amp), 100; chloramphenicol (Cam), 25; kanamycin (Kan), 50. MC4100 cells were grown to steady state (Vischer *et al*, 2015) in glucose minimal medium containing 6.33 g of $K_2HPO_4.3H_2O$, 2.95 g of $KH_2PO_4$, 1.05 g of $(NH_4)_2$, 0.10 g of $MgSO_4.7H_2O$, 0.28 mg of $FeSO_4.7H_2O$, 7.1 mg of $Ca(NO_3)_2.4H_2O$, 4 mg of thiamine and 4 g of glucose. For strain MC4100, 50 μg lysine per liter was added. Absorbance was measured at 450 nm with a 300-T-1 spectrophotometer (Gilford Instrument Laboratories Inc.). A list of all strains and plasmids used in this study can be found in Appendix Tables S1 and S2, respectively.

### Bacterial strain construction

BW25113 was used as the parent strain (WT) for this study unless otherwise stated. Strains were generated, by transducing P1 lysates derived from the corresponding deletion strains of the Keio and Aska strain collections (Adams & Luria, 1958; Baba *et al*, 2006). A list of all primers used in this study can be found in Appendix Table S3.

### Generation of the NlpI-HA-tagged strain

For HA-tagging, pKD13 (kanamycin resistant) was used as a PCR template. The kanamycin cassette was amplified by PCR with the primers 74-NlpI-HA-O1 and 87-NlpI-HA-O2. The primer 74-NlpI-HA-O1 was carrying from 5′ to 3′: the homology region of the C-terminal

of NlpI (without the STOP codon), 2 HA-tag and the homology region of the N-terminus of the kanamycin cassette (from the pKD13). The primer 87-NlpI-HA-O2 was carrying from 5′ to 3′ the homology region of the downstream region of NlpI and the homology region of the C-terminus of the kanamycin cassette.

### Generation of the NlpI-Strep-Flag (-SF) tagged strain

For SF-tagging, pJSP1 (containing the SF-tag and a kanamycin cassette) was used as a PCR template. The SF-tag and kanamycin cassette were amplified by PCR with the primers 175-NlpI-SF-O1 and 176-NlpI-SF-O2. The primer 175-NlpI-SF-O1 was carrying from 5′ to 3′ the homology region of the C-terminal of NlpI (without the stop codon) and the homology region of the N-terminal of the Strep-Flag tag (from the pJSP1). The primer 176-NlpI-SF-O2 was carrying from 5′ to 3′ the homology region of the downstream region of NlpI and the homology region of the C-terminus of the kanamycin cassette.

Transformation and antibiotic resistance selection were performed as previously described (Datsenko & Wanner, 2000). BW25113 transformants carrying a Red helper plasmid were grown in 5-ml SOB cultures with ampicillin and L-arabinose at 30°C to an $OD_{600}$ of ≈0.6 and then made electrocompetent by concentrating 100-fold and washing three times with ice-cold 10% glycerol. PCR products were gel-purified, digested with *Dpn*I, re-purified and suspended in elution buffer (10 mM Tris, pH 8.0). Electroporation was done by using a Cell-Porator with a voltage booster and 0.15-cm chambers according to the manufacturer's instructions (GIBCO/BRL) by using 25 μl of cells and 10–100 ng of PCR product. Shocked cells were added to 1 ml SOC and incubated 1 h at 37°C, and then, one-half of the incubation/cells were spread onto agar to select $Km^R$ transformants.

### Eliminating antibiotic resistance gene for the NlpI-HA

Antibiotic resistance was eliminated as described (Datsenko & Wanner, 2000). The pCP20 plasmid has ampicillin and chloramphenicol resistance genes and shows temperature-sensitive replication and thermal induction of FLP synthesis (Cherepanov & Wackernagel, 1995). $Km^R$ mutants were transformed with pCP20, and ampicillin-resistant transformants were selected at 30°C, after which a few were colony-purified once non-selectively at 43°C and then tested for loss of all antibiotic resistances. The majority of the mutants lost the FRT-flanked resistance gene and the FLP helper plasmid simultaneously.

### Immunolabelling

The specificity of the antibody was confirmed by labelling a WT and Δ*nlpI* strain with affinity-purified anti-NlpI. Quantitative analysis of the fluorescence found in the Δ*nlpI* strain gave the same level as WT cells immunolabelled with secondary antibodies only, whereas the WT cells showed a much higher fluorescence level with the purified anti-NlpI and a regular distribution of foci in the envelope.

After reaching steady state, the cells were fixed for 15 min by addition of a mixture of formaldehyde (f.c. 2.8%) and glutaraldehyde (f.c. 0.04%) to the shaking water bath and immunolabelled as described (Buddelmeijer *et al*, 2013) with rabbit polyclonal

antibodies against NlpI or against the HA-tag. As secondary antibody, donkey anti-rabbit conjugated to Cy3 or conjugated to Alexa488 (Jackson Immunochemistry, USA) diluted 1:300 in blocking buffer (0.5% (wt/vol) blocking reagents (Boehringer, Mannheim, Germany) in PBS) was used, and the samples were incubated for 30 min at 37°C. For immunolocalization, cells were immobilized on 1% agarose in water slabs coated object glasses as described (Koppelman *et al*, 2004) and photographed with an Orca Flash 4.0 (Hamamatsu) CCD camera mounted on an Olympus BX-60 fluorescence microscope through a 100×/N.A. 1.35 oil objective. Images were taken using the program ImageJ with MicroManager (https://www.micro-manager.org).

SIM images were obtained with a Nikon Ti Eclipse microscope and captured using a Hamamatsu Orca-Flash 4.0 LT camera. Phase-contrast images were acquired with a Plan APO 100×/1.45 Ph3 oil objective. SIM images were obtained with a SR APO TIRF 100×/1.49 oil objective, using 3D-SIM illumination with a 488 nm laser, and were reconstructed with Nikon-SIM software using the values 0.23–0.75–0.10 for the parameters Illumination Modulation Contrast (IMC), High Resolution Noise Suppression (HNS) and Out of focus Blur Suppression (OBS).

### Image analysis

Phase-contrast and fluorescence images were combined into hyperstacks using ImageJ (http://imagej.nih.gov/ij/), and these were linked to the project file of Coli-Inspector running in combination with the plugin ObjectJ (https://sils.fnwi.uva.nl/bcb/objectj/). The images were scaled to 15.28 pixels per μm. The fluorescence background has been subtracted using the modal values from the fluorescence images before analysis. Slight misalignment of fluorescence with respect to the cell contours as found in phase contrast was corrected using Fast-Fourier techniques as described in Vischer *et al* (2015). Data analysis was performed as described in Vischer *et al* (2015). In brief, midcell was defined as the central part of the cell comprising 0.8 μm of the axis. From either cell part, midcell or remaining cell, the volume, the integrated fluorescence, and, thus, the concentration of fluorophores were calculated. The difference of the two concentrations is multiplied with the volume of midcell. It yields FCPlus (surplus of fluorescence). For age calculation, all cell lengths are sorted in ascending order. Then the equation:

$$age = \ln(1 - 0.5 * rank/(nCells - 1))/\ln(0.5)$$

is used, where *rank* is a cell's index in the sorted array, *nCells* is the total amount of cells, and *age* is the cell's age expressed in the range 0–1.

### Ni²⁺-NTA pull-down assay

His-tagged proteins of interest were incubated with untagged or native ligands, in the presence of $Ni^{2+}$-NTA-coupled agarose beads (Qiagen). Beads were pre-equilibrated with $dH_2O$ and binding buffer (10 mM HEPES/NaOH, 10 mM $MgCl_2$, 150 mM, NaCl 0.05% Triton X-100, pH 7.5) by centrifugation at 4,000 *g*, 4 min at 4°C. Samples were incubated overnight on a spinning plate at 4°C before beads were washed 3–6 times with 10 mM HEPES/NaOH, 10 mM $MgCl_2$,

150 mM, NaCl 0.05% Triton X-100, 30 mM imidazole, pH 7.5. Retained material was eluted from $Ni^{2+}$-NTA beads using proteus spin columns and boiling at 100°C in SDS-buffer (50 mM Tris–HCl pH 6.8, 2% SDS, 10% glycerol 0.02% bromophenol blue, 10% β-mercaptoethanol). Elutions were diluted 1:1 with $dH_2O$, and proteins were separated by SDS–PAGE for analysis.

### Protein overexpression and purification

Prior to purification, plasmids of interest were transformed into *E. coli* strain BL21 (λDE3) and grown overnight in LB agar (1.5% w/v) containing appropriate antibiotic, at 37°C. Transformants were inoculated into 50 ml of LB with appropriate antibiotic and grown overnight at 37°C, shaking. Pre-cultures were diluted 1:40 in 2 l LB and grown to $OD_{578}$ 0.5–0.6, at 37°C. Induction conditions are specified for each respective protein below. After overexpression, cells were harvested by centrifugation at 7,500 *g*, 15 min, 4°C. Pellets were re-suspended in buffer I (25 mM Tris–HCl, 300 mM NaCl, pH 7.5) with the addition of a small amount of DNase (Sigma) and 100 μM P.I.C and PMSF. Cells were lysed by sonication (Branson digital) and the lysate was centrifuged at 14,000 *g*, 1 h, 4°C, before the supernatant was applied at 1 ml/min to a 5 ml chromatography column attached to an ÄKTA Prime plus (GE Healthcare).

If desired, the removal of his-tags for tagged constructs, following immobilized metal affinity chromatography steps, was achieved by incubating protein samples with 1 unit/ml of restriction grade thrombin (Novagen). This was carried out overnight at 4°C in 25 mM Tris–HCl, 200 mM NaCl, pH 8.0 or 25 mM HEPES/NaOH, 300 mM NaCl, 10% glycerol, pH 7.5, depending on the next purification step. Removal of His-tag was verified by Western blot with monoclonal α-His–HRP (1:10000) antibody (Sigma).

### Purification of MepM

MepM was purified as previously described in Moré *et al* (2019).

### Purification of MepS and MepS(C68A)

MepS and MepS$^{C68A}$ (MepS*) overexpression was induced with 1 mM IPTG for 90 min at 37°C. Following harvesting, lysate was applied to a 5 ml HisTrap HP column (GE Healthcare) in buffer containing 25 mM Tris–HCl, 300 mM NaCl, 20 mM imidazole pH 7.5. Protein was eluted in 25 mM Tris–HCl, 300 mM NaCl, 400 mM imidazole, 10% glycerol, pH 7.5. Protein purity and yield were analysed by SDS–PAGE, and the fractions of interest were pooled and dialysed overnight against 25 mM HEPES/NaOH, 300 mM NaCl, 10% glycerol, pH 7.5. Protein was concentrated to ~5 ml using Vivaspin concentrator spin columns (Sartorius) at 4,500 *g*, 4°C and applied to a HiLoad 16/600 Superdex 200 column (GE Healthcare) at 1 ml/min. Protein purity and yield were analysed by SDS–PAGE, and the best fractions were pooled and stored at −80°C. Disclaimer: The authors note that the purification of an "active" preparation of MepS from pET21b-MepS-His (Singh *et al*, 2012) was difficult and irreproducible. We were not able to purify an "active" version of MepS to show activity against muropeptides on its own; however, we were able to consistently detect low levels of activity in the presence of NlpI. We addressed NlpI stimulation using this "active" preparation of MepS in the manuscript. However, subsequent

purifications of MepS were not always consistent in showing this stimulation by NlpI.

### Purification of NlpI

NlpI overexpression was induced with 1 mM IPTG, 3 h at 30°C, before harvesting of cells as described above. Following harvesting, lysate was applied to a 5 ml HisTrap HP column (GE Healthcare) and washed with buffer containing 25 mM Tris–HCl, 300 mM NaCl, 20 mM imidazole pH 7.5. Protein was eluted in 25 mM Tris–HCl, 300 mM NaCl, 400 mM imidazole, 10% glycerol, pH 7.5. Protein purity and yield were analysed by SDS–PAGE, and the fractions of interest were pooled and dialysed overnight against 25 mM HEPES/NaOH, 200 mM NaCl, 10% glycerol, pH 7.5. Protein was concentrated to ~ 5 ml using Vivaspin concentrator spin columns (Sartorius) at 4,500 *g*, 4°C and applied to a HiLoad 16/600 Superdex 200 column (GE Healthcare) at 1 ml/min. Protein purity and yield were analysed by SDS–PAGE, and the best fractions were pooled and stored at −80°C.

### Purification of PBP4

Purification of native PBP4 followed an adapted protocol from Kishida *et al* (2006). PBP4 overexpression was induced with 1 mM IPTG for 8 h at 20°C and then harvested by centrifugation at 7,500 *g*, 4°C, 15 min. Cell pellets were re-suspended in buffer (50 mM Tris–HCl, 300 mM NaCl, pH 8.0) and lysed by sonication before centrifuging at 14,000 *g*, 1 h, 4°C and reducing NaCl concentration by stepwise dialysis in a Spectra/Por dialysis membrane (MWCO 12–14 kDa). Cell supernatant was first dialysed against dialysis buffer I (50 mM Tris–HCl, 200 mM NaCl, pH 8.5) for 1 h at 4°C, then against dialysis buffer II (50 mM Tris–HCl, 100 mM NaCl, pH 8.5) for 1 h at 4°C and then finally against dialysis buffer III (50 mM Tris–HCl, 30 mM NaCl, pH 8.5), O/N at 4°C. Dialysed protein sample was then centrifuged at 7,500 *g*, 4°C, 10 min, and supernatant applied to a 5 ml HiTrap Q HP IEX column in 25 mM Tris–HCl, 30 mM NaCl, pH 8.5. Protein was eluted from the column with a linear gradient of buffer 2 containing 25 mM Tris–HCl, 1 M NaCl, pH 8.0, over a 100 ml volume. Fractions of interest were analysed by SDS–PAGE, and the best fractions were pooled and dialysed O/N, at 4°C, against dialysis buffer containing 10 mM potassium phosphate, 300 mM NaCl, pH 6.8. Protein was applied at 1 ml/min to a 5 ml ceramic hydroxyapatite column (Bio-Rad Bioscale™) in the dialysis buffer. Fractionation of proteins was achieved by using a linear gradient of buffer 2 (500 mM potassium phosphate, 300 mM NaCl, pH 6.8) over a 50-ml gradient. Fractions of highest purity and yield were dialysed overnight against 25 mM HEPES/NaOH, 300 mM NaCl, 10% glycerol, pH 7.5 and concentrated to ~ 5 ml using Vivaspin concentrator spin columns (Sartorius). Protein sample was applied to a HiLoad 16/600 Superdex 200 column (GE Healthcare) at 1 ml/min pre-equilibrated with $dH_2O$ and buffer I (25 mM HEPES/NaOH, 300 mM NaCl, 10% glycerol, pH 7.5). Samples were analysed by SDS–PAGE, and fractions containing purified protein were pooled and stored at −80°C.

### Purification of PBP7

PBP7 overproduction was induced with 1 mM IPTG for 3 h at 30°C before being harvested by centrifugation as described above and re-

suspended in buffer I (25 mM Tris–HCl, 500 mM NaCl, 20 mM imidazole, pH 7.5). Following sonication and subsequent centrifugation as described above, the lysate was applied to a 5 ml HisTrap HP column (GE Healthcare) and washed with four column volumes of buffer I; before bound protein was eluted with buffer II (25 mM Tris–HCl, 300 mM NaCl, 400 mM imidazole pH 7.5). Samples were analysed by SDS–PAGE and dialysed overnight against 25 mM HEPES/NaOH, 300 mM NaCl, 10% glycerol, pH 7.5, before being concentrated to ~5 ml using Vivaspin concentrator spin columns (Sartorius) at 4,500 $g$, 4°C. Protein samples were applied to a HiLoad 16/600 Superdex 200 column (GE Healthcare) at 1 ml/min pre-equilibrated with dH$_2$O and buffer I (25 mM HEPES/NaOH, 300 mM NaCl, 10% glycerol, pH 7.5). Samples were analysed by SDS–PAGE, and the purest fractions with highest yield were pooled and stored at −80°C.

### Purification of PBP1A and LpoA

Purification of PBP1A and LpoA was as described previously in Born et al (2006) and Jean et al (2014), respectively.

### Microscale Thermophoresis assays

NlpI, MepS, PBP1A, PBP4 and PBP7 were labelled on amines with NT647 RED-N-hydroxysuccinimide (NHS) reactive dye, whilst LpoA was labelled on cysteines with NT647 RED-Maleimide reactive dye (Nanotemper) according to manufacturer's instructions (Nanotemper) and as described in Jerabek-Willemsen et al (2011).

Two-fold serial dilution of proteins was done in MST buffer containing 25 mM HEPES/NaOH, 150 mM NaCl, 0.05% Triton X-100, pH 7.5. Unlabelled ligands were titrated from the following starting concentrations: NlpI (50 μM), MepM (30 μM), MepS(C68A) (30 μM), PBP4 (50 μM), PBP7 (30 μM) and LpoA (30 μM). Ligands were serially diluted 16 times and assayed for interactions with labelled proteins of interest at 10–40% MST power in standard or premium capillaries on a Monolith NT.115. Binding curves and kinetic parameters were plotted and estimated using NT Analysis 1.5.41 and MO. Affinity Analysis (x64) software.

### SDS Denaturation (SD) test

Prior to all MST measurements, capillary scans were carried out to check for consistent initial fluorescence counts. In assays that showed concentration-dependent changes in fluorescence intensity, SDS denaturation tests were carried out to investigate whether changes were non-specific or were a property of ligand binding. 10 μl from samples containing the highest and lowest concentration of unlabelled ligand was centrifuged 10,000 $g$, 5 min, RT and mixed 1:1 volume ratio with SD-test buffer (40 mM DTT, 4% SDS). Mixtures were boiled at 100°C for 10 min to abolish ligand binding before being spun down and subjected to another capillary scan. If fluorescence intensities were back to within the margin of error, then initial changes were due to ligand binding and binding curves were plotted using raw fluorescence values.

### Fixed ligand concentration MST assays for trimeric complexes

For fixed ligand concentration MST assays, labelled proteins were titrated against a fixed concentration of unlabelled proteins, respectively. In fixed concentration assays with labelled MepS or LpoA, unlabelled PBP4-NlpI or PBP7-NlpI complexes were pre-formed by incubating NlpI (3 μM) with excess PBP4 or PBP7 (30 μM), on ice for 10 min. In fixed concentration assays with labelled PBP1A, unlabelled PBP4-NlpI complex was pre-formed by incubating PBP4 (0.5 μM) with NlpI (1 μM), on ice for 10 min. Thermophoresis or fluorescence of labelled protein in the presence of unlabelled ligands was determined using NT Analysis 1.5.41 and MO Affinity Analysis (×64) software.

### Analytical ultracentrifugation

Purified NlpI was dialysed O/N against 25 mM HEPES/NaOH, 150 mM NaCl, pH 7.5, in preparation for AUC. AUC sedimentation velocity (SV) experiments were carried out in a Beckman Coulter (Palo Alto, CA, USA) ProteomeLab XL-I analytical ultracentrifuge using absorbance at 280 nm and interference optics. All AUC runs were carried out at a rotation speed of 45,000 rpm at 20°C using an 8-hole AnTi50 rotor and double-sector aluminium-Epon centre-pieces. The sample volume was 400 μl and the sample concentrations ranged between 0.3 and 1.2 mg/ml. The partial specific volumes ($\bar{v}$) for the proteins were calculated from the amino acid sequence of NlpI, using the program SEDNTERP (Laue et al, 1992). Sedimentation velocity profiles were treated using the size-distribution c(s) model implemented in the program SEDFIT14.1 (Schuck, 2000). The experimental values of the sedimentation coefficient were corrected for the viscosity and density of the solvent, relative to that of H$_2$O at 20°C ($s_{20,w}$). The atomic coordinates from the published crystal structure of (Wilson et al, 2005) were used to calculate the sedimentation coefficient values for the monomer and dimer of NlpI using the program SoMo (Brookes et al, 2010).

### In vitro PG digestion assays

PG digestion assays and subsequent muropeptide composition analysis were carried out as previously described (Glauner, 1988). 10% (v/v) substrate isolated from E. coli strain MC1061 was utilized in digestion reactions as follows: MepM (2 μM) ± NlpI (4 μM) incubated against intact sacculi for 4 h, MepS (5 μM) ± NlpI (10 μM) incubated against muropeptides O/N, PBP4 (2 μM) ± NlpI (4 μM) incubated against sacculi for 4 h, PBP7 (2 μM) ± NlpI (4 μM) incubated against muropeptides for 4 h. Standard reaction conditions were 10 mM HEPES/NaOH, 10 mM MgCl$_2$, 150 mM NaCl, 0.05% Triton X-100, pH 7.5, in 100 μl reaction volume. Following incubation, samples were boiled at 100°C, 10 min, to terminate reactions before digesting remaining PG overnight at 37°C, with 1 μM cellosyl (Hoechst, Frankfurt am Main, Germany). The samples were centrifuged at 10,000 $g$ for 5 min, RT, to obtain digested muropeptide products in the supernatant. Following digestion, muropeptide products were reduced with NaBH$_4$, adjusted to pH 4–5 and separated for analysis by reversed-phase HPLC (Glauner, 1988).

Pre-digested muropeptides were obtained by incubating intact sacculi from E. coli strain MC1061, with 1 μM cellosyl at 37°C, overnight. Following this, reactions were terminated by boiling at 100°C for 5 min and the muropeptide substrates obtained by centrifugation at 10,000 $g$ for 5 min, RT and taking the supernatant. Reactions were then carried out and prepared for analysis as described above.

### Continuous fluorescence glycosyltransferase (GTase) assay

Dansylated lipid II was prepared as previously published (Breukink *et al*, 2003). Continuous fluorescence GTase assays were performed as described (Banzhaf *et al*, 2012), using PBP1A (final concentration 0.5 μM), LpoA (1 μM), of MepS^C68A (MepS*) (2 μM) and of NlpI (4 μM), in 50 mM HEPES/NaOH pH 7.5, 150 mM NaCl, 25 mM MgCl$_2$, 0.05% Triton X-100. Briefly, dansylated lipid II was added to start the reactions and the decrease in fluorescence was measured over time at 30°C using a plate reader (excitation wavelength of 330 nm, emission of 520 nm).

### PG GTase activity assay

Substrate was prepared for the assay as follows: 0.5 μM ATTO$^{647}$-labelled lipid II was mixed with 25 μM unlabelled lipid II in 1:1 chloroform-methanol. The mixture was dried and re-suspended in 0.2% Triton X-100. Reactions were carried out in the presence of 1 mM ampicillin in 10 mM HEPES/NaOH, 10 mM MgCl$_2$, pH 7.5, 150 mM NaCl, 0.05% Triton X-100. Samples were incubated for 1 h at 37°C and boiled at 100°C for 5 min, to terminate reactions. Samples (15 μl) were dried in a vacuum concentrator before being re-dissolved in 4 μl of loading buffer (60 mM Tris–HCl, pH 8.8, 25% glycerol, 2% SDS + bromophenol blue). Glycan chain products were analysed by Tris-Tricine SDS–PAGE (Meeske *et al*, 2016; Egan *et al*, 2018).

### Measurement of TPase activity using radiolabelled lipid II

Measurement of TPase activity using [$^{14}$C]GlcNAc-labelled lipid II substrate was carried out as previously described (Bertsche *et al*, 2005). Lipid II stored in chloroform/methanol (1.2 nM) was vacuum dried in glass tubes and re-suspended in 5 μl 0.2% Triton X-100. The reactions were carried out using PBP1A (0.5 μM), LpoA$^{sol}$ (1 μM), MepS* (2 μM), NlpI$^{sol}$ (4 μM), as required, in 10 mM HEPES/NaOH, 100 mM MgCl$_2$, 150 mM NaCl, 0.05% Triton X-100 in a final volume of 100 μl. Reactions were initiated by adding the reaction mixtures to the substrate and then incubating at 37°C for 1.5 h with shaking. Reactions were terminated by boiling at 100°C for 10 min, before samples were adjusted to pH 4.8 and incubated with ~1 μM cellosyl (Hoechst, Germany) for a further 1.5 h at 37°C. Samples were then boiled at 100°C for 10 min and muropeptides reduced with NaBH$_4$ (in 0.5 M sodium borate buffer), prior to HPLC analysis as described in Glauner (1988).

### Purification of anti-NlpI

This protocol was adapted from a previously published method (Banzhaf *et al*, 2012). Serum against NlpI was obtained from rabbits at Eurogentec (Herstal, Belgium), using purified oligohistidine-tagged NlpI protein for immunization. For affinity purification of the serum, purified His-NlpI (5 mg) was coupled to 0.45 g of CNBr-activated sepharose (GE) following the manufacturers protocol. The column was washed with 30 ml of wash buffer I (10 mM Tris–HCl, 1 M NaCl, 10 mM MgCl2, 0.1% Triton X-100, pH 7.2), 5 ml of elution buffer I (100 mM Glycine/HCl, 0.1% Triton X-100 pH 2.0) and equilibrated with 30 ml of block buffer (200 mM Tris–HCl, 500 mM NaCl, 0.1% Triton X-100 pH 8.0). Rabbit serum (10 ml)

was mixed with 35 ml of serum buffer (10 mM Tris–HCl pH 7.4) and adjusted to a total concentration of 0.1% of Triton X-100. The solution was centrifuged (4,000 *g*, 45 min, 4°C), and the supernatant was applied to the CNBr-activated sepharose His-PBP2 column using a peristaltic pump with constant slow flow for 48 h. The column was washed with 30 ml of wash buffer I and with 20 ml of wash buffer II (10 mM Tris–HCl, 150 mM NaCl, 10 mM MgCl$_2$, 0.1% Triton X-100, pH 7.2). The NlpI antibodies were eluted with 10 ml of elution buffer I and mixed with 2 ml of elution buffer II (2 M Tris–HCl, pH 8.0) afterwards. The elution was analysed by SDS–PAGE, and glycerol was added to a final concentration of 20% and the purified PBP2 antibodies were stored at −20°C. Anti-NlpI was tested for specificity by Western blot (Appendix Fig S5C).

### Preparation of membrane fraction for affinity chromatography

This protocol was adapted from a previously published method (Vollmer *et al*, 1999). Membranes were isolated from 4 l of *E. coli* BW25133 grown at 37°C to an optical density (578 nm) of 0.7. Cells were harvested at (5,000 *g*, 10 min, 4°C), re-suspended in 20 ml of MF buffer I (10 mM Tris/maleate, 10 mM MgCl$_2$, pH 6.8) and disrupted by sonication, with a Branson Digital Sonifier operating at 50 W for 5 min. Membranes were sedimented by ultracentrifugation (80,000 *g*, 60 min, 4°C). The pellet was re-suspended in 20 ml of MF buffer II (10 mM Tris–maleate, 10 mM MgCl$_2$, 1 M NaCl, 2% Triton X-100, pH 6.8) to extract all membrane proteins by stirring overnight at 4°C. The supernatant obtained after another ultracentrifugation step (80,000 *g*, 60 min, 4°C) was diluted by the addition of 20 ml of MF dialysis buffer I (10 mM Tris/maleate, 10 mM MgCl$_2$, 50 mM NaCl, pH 6.8) and dialysed against 5 l of the same buffer. The obtained membrane fraction was used directly for affinity chromatography. For high salt affinity chromatography, the obtained membrane fraction was dialysed against 3 l of MF buffer III (10 mM Tris/maleate, 10 mM MgCl$_2$, 400 mM NaCl, pH 6.8). For membrane extracts using the detergent DDM, Triton X-100 was replaced with 1% DDM.

### Affinity chromatography

This protocol was adapted from a previously published method (Vollmer *et al*, 1999). Sepharose beads were activated following the instructions of the manufacturer (GE). Coupling of 2 mg of protein to 0.13 g of activated sepharose beads was carried out overnight at 6°C with gentle agitation in protein buffer. After washing the beads with protein buffer, the remaining coupling sites were blocked by incubation in AC blocking buffer (200 mM Tris–HCl, 10 mM MgCl$_2$, 500 mM NaCl, 10% glycerol and 0.25% Triton X-100, pH 7.4) with gentle agitation overnight at 6°C. The beads were washed alternatingly with AC blocking buffer and AC acetate buffer (100 mM sodium acetate, 10 mM MgCl$_2$, 500 mM NaCl, 10% glycerol and 0.25% Triton X-100, pH 4.8) and finally re-suspended in AC buffer I (10 mM Tris/maleate, 10 mM MgCl$_2$, 50 mM NaCl, 1% Triton X-100, pH 6.8). As control (Tris-Sepharose), one batch of activated Sepharose beads was treated identically with the exception that no protein was added. Affinity chromatography was performed at 6°C. *E. coli* membrane fraction extracted out of 2 l per sample (see above) containing 50 mM NaCl (or 400 mM NaCl for high salt chromatography) was incubated with gentle agitation overnight. The

column was washed with 10 ml of AC wash buffer (10 mM Tris/maleate, 10 mM MgCl$_2$, 50 mM NaCl and 0.05% Triton X-100, pH 6.8). Retained proteins were eluted with 20 ml of AC elution buffer I (10 mM Tris/maleate, 10 mM MgCl$_2$, 150 mM NaCl, 0.05% Triton X-100, pH 6.8) followed by a second elution step with 1 ml of AC elution buffer II (10 mM Tris/maleate, 10 mM MgCl$_2$, 1 M NaCl, 0.05% Triton X-100, pH 6.8). Both elution fractions were stored at −20°C. For the high salt affinity chromatography, the AC high salt wash buffer (10 mM Tris/maleate, 10 mM MgCl$_2$, 400 mM NaCl and 0.05% Triton X-100, pH 6.8) and the AC high salt elution buffer (10 mM Tris/maleate, 10 mM MgCl$_2$, 2 M NaCl, 0.05% Triton X-100, pH 6.8) were used. Elutions were analysed by liquid chromatography (LC)-MS/MS.

### Mass spectrometry to identify NlpI affinity chromatography hits

For liquid chromatography (LC)-MS/MS, tryptic peptides were desalted (Oasis HLB μElution Plate, Waters), dried in vacuum and reconstituted in 20 μl of 4% acetonitrile, 0.1% formic acid. In total, 1 μg of peptide was separated with a nanoACQUITY UPLC system (Waters) fitted with a trapping column (nanoAcquity Symmetry C$_{18}$; 5 μm [average particle diameter]; 180 μm [inner diameter] × 20 mm [length]) and an analytical column (nanoAcquity BEH C$_{18}$; 1.7 μm [average particle diameter]; 75 μm [inner diameter] × 200 mm [length]). Peptides were separated on a 240-min gradient and were analysed by electrospray ionization–tandem mass spectrometry on an Orbitrap Velos Pro (Thermo Fisher Scientific). Full-scan spectra from a mass/charge ratio of 300 to one of 1,700 at a resolution of 30,000 full widths at half maximum were acquired in the Orbitrap mass spectrometer. From each full-scan spectrum, the 15 ions with the highest intensity were selected for fragmentation in the ion trap. A lock-mass correction with a background ion (mass/charge ratio, 445.12003) was applied.

The raw mass spectrometry data were processed with MaxQuant (v1.5.2.8; Cox & Mann, 2008) and searched against an Uniprot *E. coli* K12 proteome database. The search parameters were as following: carbamidomethyl (C) (fixed), acetyl (N-term) and oxidation (M) (variable) were used as modifications. For the full-scan MS spectra (MS1), the mass error tolerance was set to 20 ppm and for the MS/MS spectra (MS2) to 0.5 Da. Trypsin was selected as protease with a maximum of two missed cleavages. For protein identification, a minimum of one unique peptide with a peptide length of at least seven amino acids and a false discovery rate below 0.01 were required on the peptide and protein level. The match between runs function was enabled, and a time window of one minute was set. Label-free quantification was selected using iBAQ (calculated as the sum of the intensities of the identified peptides and divided by the number of observable peptides of a protein) (Schwanhausser *et al*, 2011), with the log fit function enabled.

The proteinGroups.txt file, an output of MaxQuant, was loaded into R (ISBN 3-900051-07-0) for further analysis. The iBAQ values of the MaxQuant output were first batch-corrected using the limma package (Ritchie *et al*, 2015) and then normalized with the vsn package (Huber *et al*, 2002). Individual normalization coefficients were estimated for each biological condition separately. Limma was used again to test the normalized data for differential expression. Proteins were classified as a "hit" with a log$_2$ fold change higher than 4 and a "candidate" with a log$_2$ fold change higher than 2.

### Genetic interaction assay

For quantitation of genetic interactions, strains were grown to late exponential phase (~0.7 OD$_{578}$), adjusted to an OD$_{578}$ of 1 and spread out using glass beads on rectangular LB Lennox plates (200 μl per strain per plate). Plates were dried at 37°C for one hour and before they were used as a source plate for the genetic interaction assay. One source plate for each strain was arrayed using a Rotor HDA replicator on Lennox LB agar plates to transfer 96 clones to the genetic interaction plate. On each genetic interaction assay plate, the parental strain, the single deletion A, the single deletion B and the double deletion AB (or BA) were arrayed, each in 96 copies per plate. Plates were incubated at 37°C for 12 h and imaged under controlled lighting conditions (spImager S&P Robotics) using an 18 megapixel Canon Rebel T3i (Canon). Colony integral opacity as fitness readout was quantified using the image analysis software Iris (Kritikos *et al*, 2017). Double-mutant genetic interaction scores were calculated as previously described. Briefly, fitness ratios are calculated for all mutants by dividing their fitness values by the respective WT fitness value. The product of single mutant fitness ratios (expected) is compared to the double mutant fitness ratio (observed) across replicates. The probability that the two means (expected and observed) are equal across replicates is obtained by a Student's two-sample *t*-test.

### Thermal proteome profiling and sample preparation

Thermal proteome profiling was performed as previously described in Mateus *et al* (2018). Briefly, bacterial cells were grown overnight at 37°C in lysogeny broth and diluted 100-fold into 20 ml of fresh medium. Cultures were grown aerobically at 37°C with shaking until optical density at 578 nm (OD$_{578}$) ~0.5. Cells were then pelleted at 4,000 *g* for 5 min, washed with 10 ml PBS, re-suspended in the same buffer to an OD$_{578}$ of 10 and aliquoted to a PCR plate. The plate was subjected to a temperature gradient for 3 min in a PCR machine (Agilent SureCycler 8800), followed by 3 min at room temperature. Cells were lysed with lysis buffer (final concentration: 50 μg/ml lysozyme, 0.8% NP-40, 1× protease inhibitor (Roche), 250 U/ml benzonase and 1 mM MgCl$_2$ in PBS) for 20 min, shaking at room temperature, followed by three freeze–thaw cycles. Protein aggregates were then removed, and the soluble fraction was digested according to a modified SP3 protocol (Mateus *et al*, 2018). Peptides were labelled with TMT6plex (Thermo Fisher Scientific), desalted with solid-phase extraction on a Waters OASIS HLB μElution Plate (30 μm) and fractionated onto six fractions on a reversed-phase C18 system running under high pH conditions.

### 2D-TPP mass spectrometry-based proteomics

Samples were analysed with liquid chromatography coupled to tandem mass spectrometry, as previously described (Mateus *et al*, 2018). Briefly, peptides were separated using an UltiMate 3000 RSLC nano-LC system (Thermo Fisher Scientific) equipped with a trapping cartridge (Precolumn C18 PepMap 100, 5 μm, 300 μm i.d. × 5 mm, 100 Å) and an analytical column (Waters nanoEase HSS C18 T3, 75 μm × 25 cm, 1.8 μm, 100 Å). Solvent A was 0.1% formic acid in LC-MS grade water, and solvent B was 0.1% formic acid in LC-MS grade acetonitrile. After loading the peptides onto the trapping cartridge (30 μl/min of solvent A for 3 min), elution was performed

with a constant flow of 0.3 μl/min using a 60–120 min analysis time (with a 2–28% B elution, followed by an increase to 40% B, and re-equilibration to initial conditions). The LC system was directly coupled to a Q Exactive Plus mass spectrometer (Thermo Fisher Scientific) using a Nanospray-Flex ion source and a Pico-Tip Emitter 360 μm OD × 20 μm ID; 10 μm tip (New Objective). The mass spectrometer was operated in positive ion mode with a spray voltage of 2.3 kV and capillary temperature of 320°C. Full-scan MS spectra with a mass range of 375–1,200 m/z were acquired in profile mode using a resolution of 70,000 (maximum fill time of 250 ms or a maximum of 3e6 ions (automatic gain control, AGC)). Fragmentation was triggered for the top 10 peaks with charge 2–4 on the MS scan (data-dependent acquisition) with a 30-s dynamic exclusion window (normalized collision energy was 32), and MS/MS spectra were acquired in profile mode with a resolution of 35,000 (maximum fill time of 120 ms or an AGC target of 2e5 ions).

### 2D-TT data analysis

#### Protein identification and quantification

Mass spectrometry data were processed as previously described (Mateus *et al*, 2018). Briefly, raw mass spectrometry files were processed with IsobarQuant (Franken *et al*, 2015) and peptide and protein identification were performed with Mascot 2.4 (Matrix Science) against the *E. coli* (strain K12) Uniprot FASTA (Proteome ID: UP000000625), modified to include known contaminants and the reversed protein sequences (search parameters: trypsin; missed cleavages 3; peptide tolerance 10 ppm; MS/MS tolerance 0.02 Da; fixed modifications were carbamidomethyl on cysteines and TMT10-plex on lysine; variable modifications included acetylation on protein N-terminus, oxidation of methionine and TMT10plex on peptide N-termini).

#### Thermal proteome profiling analysis

Data analysis was performed in R, as previously described in Mateus *et al* (2018). Briefly, all output data from IsobarQuant were normalized using variance stabilization (*vsn*) (Huber *et al*, 2002). Abundance and stability scores were calculated with a bootstrap algorithm (Becher *et al*, 2018), together with a local FDR that describes the quality and the reproducibility of the score values (by taking into account the variance between replicates). A local FDR < 0.05 and a minimum absolute score of 10 were set as thresholds for significance. Abundance and stability scores of knocked out genes were discarded.

### CPRG assay

The CPRG assay was performed as described in Paradis-Bleau *et al* (2014). Strains were transformed with the plasmid pCB112 encoding β-galactosidase (LacZ) and grown for 16 h on CPRG medium LB [75 mM NaCl agar supplemented with CPRG (20 μg/ml), chloramphenicol (20 μg/ml) and IPTG (50 μM)] prior taking an end-point picture. CPRG (yellow) conversion to CPR (red) indicates impaired envelope integrity.

### MepS Western blot

Concerning the MepS polyclonal antibody used the Western blot, the peptide CMGKSVSRSNLRTGD, corresponding to the amino acids 120–133 of MepS, was synthetized by Proteogenix (Schiltigheim, France) and used for immunization and primary antibody generation in rabbits. The antibody was further purified in affinity column, against the antigen.

Cultures of 100 ml were grown at 30°C until an OD$_{600}$≈0.4. Cells were collected by centrifugation (3,260 *g* for 15 min at 4°C), and the pellets were suspended in 10 ml of lysis buffer (50 mM Tris–HCl pH 7.5, 100 mM NaCl and a protease inhibitor cocktail (Thermo Scientific)) and flash-frozen in liquid nitrogen. After a soft thawing on ice, cells were disrupted by sonication (Sonicator Fisherbrand FB120) (alternating 3 cycles of 30-s ON with 40% amplitude and 15-s OFF to cool down the sample). The lysates were centrifuged (3,260 *g* for 5 min at 4°C) to remove unbroken cells. The supernatant was collected and centrifuged for 1 h at 90,000 *g* at 4°C. After ultracentrifugation, the supernatant corresponds to soluble material and the pellet contains the membrane fraction. The pellet was suspended in 200 μl of lysis buffer. Protein concentration was determined using a Bradford-based colorimetric assay (Bio-Rad 5000006) (Bradford, 1976) with known concentrations of bovine serum albumin (BSA) (Sigma) as a standard. Prior to SDS–PAGE loading, samples were diluted in 4× Laemmli sample buffer (10% β-mercaptoethanol) (Bio-Rad) and concentrations were adjusted to load 6 μg of membrane proteins per lane.

Samples were separated using SDS–PAGE using a 4–20% polyacrylamide (Mini PROTEAN TGX gel, Bio-Rad) and transferred onto PVDF membranes (Bio-Rad). Membranes were blocked by 3% milk in 1× TBS-T (Tris, NaCl, Tween-20) for 1 h at RT and then incubated overnight at 4°C with MepS primary antibody (1:1,000 in 1× TBS-T Milk 3%). Membranes were washed three times with 1× TBS-T for 5 min and incubated for 1 h at RT with the secondary antibody (Goat Anti-Rabbit, Bio-Rad) coupled with horseradish peroxidase (HRP) (1:3,000 in 1× TBS-T). Prior to signal detection, membranes were washed three times with 1× TBS-T for 5 min and overlaid with ECL prime detection reagent (GE Healthcare).

### Microscopy, cell width measurements

For phase imaging and cell shape measurements, cells were grown and collected at steady state at 30°C at OD$_{600}$≈0.1. Cells were concentrated 20 times, and 0.4 μl was transferred to a 1% agarose pad (UltraPure Agarose; Invitrogen) prepared with LB and preheated at 30°C. The pad was supplemented with Carb 100 μg/ml and L-arabinose 0.2% or glucose 0.2% if specified. Phase images were obtained with an inverted epi-fluorescence Eclipse Ti microscope (Nikon), equipped with a 100× phase contrast objective (CFI PlanApo LambdaDM100X 1.4NA, Nikon). Images were acquired using a sCMOS camera (Orca Flash 4.0, Hamamatsu, Japan) with an effective pixel size of 65 nm. Cell boundaries were detected from phase-contrast microscopy images using the MATLAB-based cell segmentation tool Morphometrics (SimTK) (Ursell *et al*, 2017).

## Data availability

The mass spectrometry proteomics data have been deposited to the ProteomeXchange Consortium via the PRIDE partner repository with the dataset identifiers PXD016825 (http://www.ebi.ac.uk/pride/arc

hive/projects/PXD016825) and PXD016819 (http://www.ebi.ac.uk/pride/archive/projects/PXD016819).

Expanded View for this article is available online.

## Acknowledgements

We thank Manjula Reddy for fruitful discussions and providing reagents. This work was supported by the Sofja Kovalevskaja Award of the Alexander von Humboldt Foundation and EMBL core funding to A.T., the Wellcome Trust (101824/Z/13/Z) and Medical Research Council (MR/N002679/1) to V.W. M.B. is funded by the Royal Society (RGS\R1\191041). A.M. is supported by a fellowship from the EMBL Interdisciplinary Postdoc (EI3POD) programme under Marie Skłodowska-Curie Actions COFUND (grant number 664726). M.W. was supported by a Humboldt postdoctoral fellowship. S.v.T received support from the European Research Council (ERC) under the European Union's Horizon 2020 research and innovation programme [Grant Agreement No. (679980)], the French Government's Investissement d'Avenir program Laboratoire d'Excellence "Integrative Biology of Emerging Infectious Diseases" (ANR-10-LABX-62-IBEID), the Mairie de Paris "Emergence(s)" programme and the Volkswagen Foundation.

## Author contributions

MB, HCLY and JV: acquisition, analysis and interpretation of data, drafting and revising the article. ST, BC, MMS, EB, AL, GK, AM, AKH, FS, MW, MP, ASS and MMS: acquisition of data, analysis and interpretation of data, revising the article. TB, AT and WV: conception and design, analysis and interpretation of data, drafting and revising the article.

## Conflict of interest

The authors declare that they have no conflict of interest.

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
