## [Review Process File · The EMBO Journal]

Outer membrane lipoprotein Nlpl scaffolds peptidoglycan hydrolases within multi-enzyme complexes in *Escherichia coli*

Manuel Banzhaf, Hamish Yau, Jolanda Verheul, Adam Lodge, George Kritikos, André Mateus, Baptiste Cordier, Ann Hov, Frank Stein, Morgane Wartel, Manuel Pazos, Alexandra Solovyova, Eefjan Breukink, Sven van Teefelen, Mikhail Savitski, Tanneke den Blaauwen, Athanasios Typas, and Waldemar Vollmer

DOI: 10.15252/embj.2019102246

Corresponding author(s): Waldemar Vollmer Contributing Authors Tanneke den Blaauwen (t.denblaauwen@uva.nl), Athanasios Typas (athanasios.typas@embl.de) have been identified as a secondary point of correspondence.

Review Timeline:

Submission Date:	15th Apr 19
Editorial Decision:	11th Jun 19
Revision Received:	7th Nov 19
Editorial Decision:	11th Dec 19
Revision Received:	20th Dec 19
Accepted:	14th Jan 20

Editor: Ieva Gailite

Transaction Report:

(Note: With the exception of the correction of typographical or spelling errors that could be a source of ambiguity, letters and reports are not edited. The original formatting of letters and referee reports may not be reflected in this compilation. Depending on transfer agreements, referee reports obtained elsewhere may or may not be included in this compilation.)

Thank you for submitting your manuscript for consideration by the EMBO Journal. I apologise for the delay in communicating the decision due to holidays here in Germany and the high manuscript submission rate to our office at the moment. We have now received three referee reports on your manuscript, which are included below for your information.

As you can see from the comments, while all referees express interest in the topic of the study, reviewers #1 and #2 raise several substantial concerns that need to be addressed in order to consider publication here. Based on the overall interest expressed in the reports, I would like to invite you to submit a revised version of your manuscript in which you address the comments of all three referees, particularly focusing on the following points:

- verification of Nlpl localisation (reviewer #1, point 1)
- further analysis of the role of Nlpl in coordination of endopeptidase activity (reviewer #2, point 2)
- further insight into the role of Nlpl in regulation of cell morphology (reviewer #2, point 3)

I should add that it is The EMBO Journal policy to allow only a single major round of revision and that it is therefore important to resolve the main concerns at this stage. I realise that the scope of the revision is rather broad, and I would understand if you were to choose not to undergo an extensive revision here. Therefore I have taken the liberty to discuss your manuscript with my colleague Andrea Leibfried, the executive editor of our sister journal Life Science Alliance, and she would be happy to offer publication after a minor revision (further insight into the more far-reaching points would not be required for publication). If you are interested in this option, Andrea would be happy to discuss the scope of the revision at any point (you can contact her at a.leibfried@life-science-alliance.org).

We generally allow three months as standard revision time. Please contact us in advance if you would need an additional extension. As a matter of policy, competing manuscripts published during this period will not negatively impact on our assessment of the conceptual advance presented by your study. However, please contact me as soon as possible upon publication of any related work to discuss how to proceed.

When preparing your letter of response to the referees' comments, please bear in mind that this will form part of the Review Process File, and will therefore be available online to the community. For more details on our Transparent Editorial Process, please visit our website.

This is a detailed and thorough study on the interactions of the outer membrane lipoprotein Nlpl and enzymes involved in peptidoglycan (PG) hydrolysis and synthesis. The overall regulation of PG hydrolases is still poorly understood and therefore the aim of this study is of general interest.

The manuscript gives detailed information on various interactions of Nlpl which are very relevant to the field. Also, the manuscript provides evidence for the first time that Nlpl may have functions during PG synthesis, not just hydrolysis. However, the real novelty of the manuscript comes from the assignment of a new function to Nlpl as a novel adaptor of PG hydrolases and the suggestion that Nlpl is involved in coordinating PG multienzyme complexes including PG synthases and hydrolases. Therefore the manuscript would be stronger if more direct evidence was given for this general role of Nlpl, such as

- Co-localization studies of Nlpl and PG synthases and hydrolases. This may be difficult to do using immunofluorescence as a tool to localize Nlpl, but should be feasible if Nlpl is fused to a fluorescence protein that folds in the periplasm, such as mNeonGreen.
- (co)- localization studies of PG synthesis/hydrolysis proteins which interact with Nlpl in the presence and absence of Nlpl
- If Nlpl has an important role in general regulation of PG synthesis and hydrolysis, one could expect some phenotype in rate of PG synthesis; rate of autolysis; overall composition of cell PG. None of this was tested in the manuscript.

Major concerns

1. Localization studies of Nlpl are not "bullet-proof". Figure S5c shows immunofluorescence data using an anti-Nlpl antibody in a WT strain and a mutant deleted for *nlpl*. Although the signal decreases in the mutant, one can still observe a spotty pattern that is present in the WT, raising questions about the specificity of the signal. It is however reassuring that data is supported by similar results obtained with anti-HA antibody in a strain expressing Nlpl-HA. How was quantification of the concentration of protein at the surface (graphs in panels c and e) done? Total fluorescence? Number of spots? This information should be given.

Given that immunofluorescence can be prone to artifacts resulting from lack of specificity of antibodies, localization data should be supported by using a fluorescent derivative of Nlpl, fused to a protein that folds in the periplasm, like mNeonGreen. This is particularly important because the authors claim that Nlpl interacts with divisome proteins (Line 158), but no enrichment of Nlpl at the septum is observed (line 262), which seems contradictory.

2. Why was the effect of Nlpl on the activity of EPases assessed using different substrates (sacculi or pre-digested mucopeptides) for different samples? Fig 3a should clearly indicate which substrate was used in which experiment, given that all data is presented in a single graph.

3. In line 251-252 authors say that the presence of more MepS decreases cell width. They indeed show that lack of MepS results in thicker cells, but that does not necessarily imply the reverse. Have authors tested the effect of an overexpression of MepS? Also, given that MepS was suggested to have a role in cell elongation, it would be useful to show measurements for cell length.

4. Was strain fitness calculated based on "colony integral opacity" (line 855) or "colony size" (Line 1251)?

5. It is surprising that fitness and morphology defects do not correlate (fig 3 b and c), as the MepS mutant and Nlpl/MepS double mutant are not severely impaired in fitness but have very thick cells.

Do authors have an explanation for this?

Minor concerns

Line 180 - It would help readers that may be less familiar with the technique, to very briefly explain the principle of MST in the context of the aim of the experiments performed.

Line 847-850 - It is not very clear how the strains are plated. Is the initial rectangular plate, inoculated with 200ul of culture (spread with glass beads) used for the Rotor HAD replicator to get the inoculum for the genetic interaction (final) plates? If so, this could be explained a bit more clearly.

Line 1206 - typo in word "hydrolases"

Line 1211 - state meaning of FDR in legend

Fig S5d - show complete western image, to enable evaluation of the specificity of antibodies.

Referee #2:

Banzhaf et al. examined the function of lipoprotein Nlpl in *E. coli* as a scaffold and regulatory protein for peptidoglycan degradation and synthesis enzymes. 2D-TT and affinity chromatography approaches suggested interactions of Nlpl with MepS, LdtF, PBP4, and EnvC. Mutations of *nlpl* or *nlpl* together with various endopeptidases result in changes in cell morphology that remain unexplained. Nlpl was added to various endopeptidases and tested for the ability to cleave peptide-crosslinked tetra-tetra peptidoglycan dimers. Nlpl inhibited endopeptidase MepM but had little or no effect on other endopeptidases in vitro. Double mutants lacking *nlpl* and a biosynthetic PBP or PBP activator showed substantial growth defects. Overall, these studies increase our understanding of protein-protein interactions in *E. coli* that relate to peptidoglycan metabolism. However, the results showing Nlpl decreasing activities of two endopeptidases are inconsistent with the model of Nlpl functioning to bring endopeptidases to a cell wall site to allow PG synthesis. Thus the reader is left with an unclear idea of Nlpl function.

1. Statistical tests for significance should be applied to the data in Fig. 2c, 2d, and 3a and that information should be supplied in the figure.
2. A major question in these studies is what the function of Nlpl is for PG degradation and synthesis. Clearly, Nlpl results in less MepS and thus less endopeptidase activity by that enzyme, and binding of MepM by Nlpl caused less dimer degradation (Fig 3a). Thus, Nlpl may function to decrease endopeptidase activities. However, the authors propose that Nlpl is functioning to bring endopeptidases to the site of PG synthesis so that they can separate PG strands (lines 453-456 and elsewhere). The data show that PG synthetic enzymes associate with Nlpl, but questions remain about the outcome of these interactions. Do the trimeric complexes involving Nlpl show increased degradation of PG dimers? Would mutants where the synthetic enzymes can't interact with the Nlpl complex show less PG synthesis, more or less endopeptidase activity, or something else? It is simply unclear at the end of these studies what the effect of Nlpl is on PG synthesis and degradation.
3. The most interesting biological results from these studies are the morphological changes and growth defects in the *nlpl* and endopeptidase mutants, but the reasons for these defects were not determined.
4. I think the idea that this report is the first example of PG synthases and hydrolases in a multi-enzyme complex with a regulatory lipoprotein (lines 310-312) is incorrect as the PBP3-AmiC-NlpD complex has been described (starting with Heidrich C et al. 2001 Mol Microbiol).

5. More information should be provided on the known interactions of E. coli endopeptidases with other PG breakdown or synthesis proteins, such as those from Höltje's studies (Romeis and Höltje JBC (1994) is one example).

6. LpoA and LpoB are specific to gamma-proteobacteria, and nplI is not found in all Gram-negative species. The authors should make it clear how widespread and conserved NplI is so that the reader will know how generally this model can be applied.

Minor points.

1. The authors should not be iffy in the discussion of their results. Lines 321, 343, and 361 all contain words that make it sound like the authors don't trust their data and don't think that the proteins actually bind one another. They measured dissociation constants, so they know the proteins bind to each other.

2. The amidase regulators and their effects on their specific interacting partners have been described, so the "appear to have specificity" phrase in lines 387-388, is an inaccurate description of what is known to occur (Uehara T et al. 2010. EMBO J).

Referee #3:

This ambitious, excellent study by Drs. Banzhaf et al. presents new, intriguing data about a fundamental question in peptidoglycan (PG) cell wall synthesis in gram-negative bacteria, using E. coli as a model. The paper focuses on the function of the outer membrane NplI protein, which previous studies had shown plays a role in targeting the MepS PG endopeptidase hydrolase for degradation by a periplasmic protease. The data in this new paper greatly extends the findings from these previous studies by showing that NplI functions in setting the cellular amount and stability of numerous other proteins, including key proteins involved in PG synthesis and hydrolytic remodeling. Numerous key findings are succinctly summarized by the headings of each section of the Results. These results are based on an extensive study using biochemical, physiological, and microscopic methods. Together, these experiments provide some of the first direct evidence in support of the Holtje model that PG synthases and hydrolases function together in a physical complex during PG synthesis. This paper strongly suggests that NplI serves as a key adaptor in coordinating complex formation between PG synthases and hydrolases in these complexes, although many details about this process remain to be addressed in future studies.

The data content in this paper is truly impressive and is based on several biochemical approaches, including high-level TMT proteomics and biochemical demonstration of tripartite complexes containing NplI and other proteins along with activity assays. These data indicate that NplI has two general functions: setting the stability of partner proteins and affecting enzymatic activity, such as inhibition of the MepM endopeptidase. Straightforward genetic experiments also indicate a nearly synthetic lethal relationship between mutations in nplI and mreB/lpoB, suggesting a requirement of NplI for function of the remaining PBP1A/LpoA synthase proteins. The large amount of data is of high quality and uniformly convincing from multiple biological replicates with appropriate statistics.

This is a well-presented, important paper that will influence the field about this central process of coordination of PG synthesis machines. There are a couple points of interpretation and presentation that if addressed, would add to this paper.

Major points

1. Lines 1, lines 102, and 459 and perhaps elsewhere. "Nucleation" has a specific biochemical meaning in the context in polymer formation that does not really apply here. Rather, NplI acts a key player in "partner switching" regulation among various PG synthesis and hydrolase proteins. Therefore, please consider changing nucleates/nucleation to "partnering" or "adapting" (e.g., in lines 459: "by way of partnering proteins in complexes containing synthases...")
2. Line 82. Does the combined data set indicate why filamentation occurs in delta-nplI mutants at higher temperatures or low osmolarity? This point might be mentioned in the Discussion.
3. Line 84. Please consider changing "hypervesiculation" to "increased membrane vesical formation".
4. Section starting on line 106 and Figure 1. The paper focuses on the effects of delta-nplI mutants on the abundance and stability of proteins involved in PG biosynthesis. However, there are many very light gray dots in Figure 1 based on the supplemental tables. That is, the delta-nplI mutation is much more pleiotropic than stressed in the paper. It is understandable to focus on PG synthesis, but this broad pleiotropy should be mentioned in the Results and commented upon in the Discussion. Can this broadly pleiotropic effect be related to other studies where PG synthesis or OM assembly was disrupted by other means?
5. Related to point 4, it should be commented on whether the nplI mutation can be cleanly complemented by ectopic expression of nplI⁺ and that the pleiotropy of the delta-nplI mutation is not due to polarity.
6. Line 123-124). Some of the abundance changes detected by proteomics are small/moderate (e.g., PBP1A, MltG, etc.). Were these smaller changes in protein amounts corroborated by quantitative western blotting?
7. Lines 156-166. Why was MepS (and MepM; line 184) not detected in the pull-down experiments? This apparently inconsistent observation is mentioned later in the Discussion, but should be mentioned here and commented upon. Is it because MepS amount is really low in the nplI⁺ bacteria due to degradation? Was the pull down shown in Figures 1C and 1D performed for the delta-nplI mutant to get at proteins whose amounts might be greatly decreased due to targeted degradation mediated by binding to NplI?
8. Related to point 4, there seem to be a lot of other proteins (more gray dots) that are pulled down by binding to NplI in Figures 1C and 1D. It seems that NplI binds to a lot of proteins. Can anything else be said about the implications of these data? Are there any other patterns that emerge?
9. Line 284-285. To a lesser extent, NplI complexes are also important to the function of PBP1B/LpoB as well. This point might be stated. Also, in line 277, was the nearly synthetic lethal relationship between delta-nplI and delta-mreB/lpoB detected in the previous lacZ-based screen by Pradis-Bleau et al? Please comment on.
10. Figure 6 and Discussion. Some of the major patterns might be pulled together even more in summary Figure 6. As noted in above, the data suggest that NplI can play two roles: affecting stability and/or abundance or changing enzymatic activity, which was assayed directly for a couple of endopeptidases. These roles might be added to Fig. 6a as two columns. Changes in relative stability and/or abundance in the absence of NplI of each protein listed can be taken from data in Figure 1a and 1b. This compilation would make it really clear that for some EPases, NplI plays a

major destabilizing role (i.e., MepS), but not others (i.e., PBP4, Fig. 1b and line 217). For the endopeptidases that were assayed, another column could indicate whether direct NplI binding affected activity or did not.

11. Methods. Based on the assays, it seems that the biochemical assays containing purified NplI were performed with solubilized NplI in the absence of outer membrane binding. This point should be qualified in the Results.

Minor points

12. Line 228. Consider changing: "on its own" to "by itself".

13. Line 256. Omit: "probably".

14. Line 262. Consider changing to: "with certain divisome proteins".

15. Line 364. Change to: "facilitates the proteolytic".

16. Line 388. Consider adding a paragraph break after: "2009)."

17. Line 390. Omit: "also".

18. Line 408. Omit: "them"

19. Line 450. Change to: ", because the latter is".

20. Section starting with line 520. Mention the antibody specificity controls done for the immunofluorescence microscopy.

21. Line 786. Comma missing before "500".

22. Tables 1, 2, and 3 could be moved to the Supplemental Materials.

Referee #1

This is a detailed and thorough study on the interactions of the outer membrane lipoprotein Nlpl and enzymes involved in peptidoglycan (PG) hydrolysis and synthesis. The overall regulation of PG hydrolases is still poorly understood and therefore the aim of this study is of general interest.

The manuscript gives detailed information on various interactions of Nlpl which are very relevant to the field. Also, the manuscript provides evidence for the first time that Nlpl may have functions during PG synthesis, not just hydrolysis. However, the real novelty of the manuscripts comes from the assignment of a new function to Nlpl as a novel adaptor of PG hydrolases and the suggestion that Nlpl is involved in coordinating PG multienzyme complexes including PG synthases and hydrolases. Therefore the manuscript would be stronger if more direct evidence was given for this general role of Nlpl, such as:

We thank the reviewer for their comments about the thoroughness and general interest of our study. We agree with the referee that additional mechanistic insights strengthen the manuscript and therefore performed new experiments and added new findings to the manuscript.

-Co-localization studies of Nlpl and PG synthases and hydrolases. This may be difficult to do using immunofluorescence as a tool to localize Nlpl, but should be feasible if Nlpl is fused to a fluorescence protein that folds in the periplasm, such as mNeonGreen.

We acquired an unpublished C-terminal Nlpl-mNeonGreen (mNG) plasmid (pDSW210torASS-Nlpl-GFP) and a strain expressing chromosomal C-terminal Nlpl-mNG (MG1665 Δ npl::pDSW210torASS-Nlpl-GFP) from Manjula Reddy's laboratory (Centre for Cellular and Molecular Biology, Habsiguda, Hyderabad, India). In addition, we constructed a plasmid expressing Nlpl-GGS-mNG from a weakened p_{trc99A} promoter. However, our results indicate that Nlpl-mNG fusions are prone to create artefacts due to proteolytic degradation. Our data is consistent with data acquired in Manjula Reddy's laboratory, which she kindly shared in personal communication with us. We address this in more detail below under Referee #1, major point one.

- (co)- localization studies of PG synthesis/hydrolysis proteins which interact with Nlpl in the presence and absence of Nlpl

Because Nlpl fusion proteins were not a viable option to perform co-localisation studies, we explored if the localisation of PG synthases is dependent on Nlpl. We decided to focus on PG synthases over PG hydrolases for two main reasons. First, both PG synthases-Nlpl

deletion strains ($\Delta nlpI\Delta mrcB$ and $\Delta nlpI\Delta mrcA$) showed severe fitness defects. Second, the main EPase that exhibited strong genetic interactions with NlpI, changes dramatically its levels in $nlpI$ cells, which makes interpretation of results very difficult.

We used a strain (AV44) that simultaneously labels both major class A PBPs with a fluorescent tag and transduced $\Delta nlpI$ to inactivate NlpI (resulting in strain B195). Using both strains, we determined the cellular localisation of PBP1A and PBP1B to investigate if PG-synthase localisation is dependent on the presence of NlpI (Figures R1 and R2, below). We did not observe any apparent differences on PBP1A and PBP1B localisation between WT and $\Delta nlpI$ cells. This could be because NlpI does not interact directly with PBP1A or PBP1B.

NlpI localisation may be affected by EPases that it is interacting with. Given the cellular redundancy of EPases and the experimental challenges to elucidate such minor changes we decided to not explore this further. Therefore, we did not add this result (Figure R1) to the current manuscript, but could add it as a supplementary figure if the Referee wishes so.

Figure R1 - Localization of msfGFP-PBP1B and mCherry-PBP1A in the presence or absence of NlpI

a. Phase contrast images and corresponding fluorescence images (msfGFP and mCherry) of AV44 strain (186::Ptet75-dcas9 , $mrcB::msfGFP-mrcB$, $mrcA::mCherry-mrcA$) and B195 (186::Ptet75-dcas9 , $mrcB::msfGFP-mrcB$, $mrcA::mCherry-mrcA$, $\Delta nlpI$ -kanR). The cells were grown at 30°C to steady-state in M63 supplemented with Glucose 0,2% and CAA 0,1%. The last panel displays an overlay of the msfGFP and the mCherry images. The scale bar equals 5 μ m.

b. Phase contrast and fluorescence images of individual cells. Left panel: strain AV44 and right panel: B195. The scale bar equals 2 μ m.

Figure R2 - Localization of msfGFP-PBP1b and mCherry-PBP1A in the presence or absence of Nlpl (single cell traces)

The graphs display the msGFP and mCherry signal intensity along the line following half of cell contour (pole to pole distance in μm). The green line corresponds to the msGFP-PBP1B signal and the red corresponds to the mCherry-PBP1a signal. For each panel, the corresponding cell is displayed as a phase contrast image and the black arrow shows the direction of the signal along the cell body. Left panels: strain AV44 (186::Ptet75-dcas9, mrcB::msfgfp-mrcB, mrcA::mcherry-mrcA) and right panels: B195 (186::Ptet75-dcas9, mrcB::msfgfp-mrcB, mrcA::mcherry-mrcA, $\Delta\text{nlpl-kanR}$). The scale bar equals 2 μm .

Material & Methods used to create Figure R1 and R2:

Cells were grown and collected at steady-state at $\text{OD}_{600} \sim 0.1$. Cells were concentrated 20 times and 0.4 μl cell culture were transferred to a 1% agarose pad (UltraPure Agarose; Invitrogen) prepared with the same growth medium and preheated at 30°C. Cells were imaged using an inverted microscope (TI-E, Nikon Inc.) equipped with a 100 Å~ phase-contrast objective (CFI PlanApo LambdaDM100Å~ 1.4NA, Nikon Inc.), a solid-state light source (Spectra X, Lumencor Inc.), a multiband dichroic (69002bs, Chroma Technology Corp.). GFP and mCherry fluorescence were measured using excitation filters (560/32 and 485/25 resp.) and emission filters (632/60 and 535/50 resp.) using the following settings: 500 ms exposure time/100% light intensity and 200 ms exposure time/100% light intensity,

respectively. Images were acquired using a sCMOS camera (Orca Flash 4.0, Hamamatsu) with an effective pixel size of 65 nm.

Strains used:

-**AV44** 186::Ptet75-dcas9, mrcB::msfgfp-mrcB, mrcA::mcherry-mrcA (published): bioRxiv 763508; doi: <https://doi.org/10.1101/763508>

-**B195** 186::Ptet75-dcas9, mrcB::msfgfp-mrcB, mrcA::mcherry-mrcA, Δ nlpI-kanR (not published). Was created transducing AV44 using the Δ nlpI phage lysate from the Keio collection.

- If NlpI has an important role in general regulation of PG synthesis and hydrolysis, one could expect some phenotype in rate of PG synthesis; rate of autolysis; overall composition of cell PG. None of this was tested in the manuscript.

We performed the suggested experiments and summarized our findings in the new Supplementary Figures S9, S10 and S11. Overall, we could not detect a significant effect of NlpI in different *in vitro* assays, and the PG composition of the *nlpI* mutant was similar to that of WT. There is the possibility that we are missing some components, or that our GTase/TPase assays with radiolabelled lipid II (Fig. 9e, S10) was not sensitive enough to detect dynamic changes. This end-point assay is the only available assay to monitor TPase; the GTase assay with dansyl-lipid II is continuous. However, as indicated in the discussion, we believe that our collective data show that the main role of NlpI is to scaffold EPases to be in close proximity with PG synthases (PG-multi enzyme complexes) and therefore the results presented in Supplementary Figures S9, S10 and S11 do not contradict our initial findings.

Further information can be found in our response to referee #2, major concern two.

Major concerns

1. Localization studies of NlpI are not "bullet-proof". Figure S5c shows immunofluorescence data using an anti-NlpI antibody in a WT strain and a mutant deleted for *nlpI*. Although the signal decreases in the mutant, one can still observe a spotty pattern that is present in the WT, raising questions about the specificity of the signal. It is however reassuring that data is supported by similar results obtained with anti-HA antibody in a strain expressing NlpI-HA.

The localization of NlpI is spotty because the protein does not move in the chemically fixed cells. This fixation step is a requirement for immunolocalization. The smooth pattern often observed using GFP is due to the movement of the molecules during the time period (usually ~330-1000 ms) required to collect the image. The purified antibodies produce the same level of fluorescence as the secondary antibodies only in the Δ nlpI strain, which shows that the

labelling with the purified antibodies is specific. Please see the Extended Experimental Procedures in the Supplementary Information of Typas *et al.* (2010) Cell 143, 1097-1109, for the validation of this immunolocalization approach.

How was quantification of the concentration of protein at the surface (graphs in panels c and e) done? Total fluorescence? Number of spots? This information should be given.

The concentration of the proteins was calculated by dividing the total fluorescence per cell by the volume of that cell, assuming that the cell is a cylinder with two half spheres as cell poles. In case of the concentration in the cell envelope per μm^2 we calculated the surface of the cells using the same model.

The calculation of the Nlpl amount as function of the cell division age was based on the conversion of the fluorescent unit to molecules based on the number of Nlpl molecules in minimal medium grown cells (1) as described in (2).

References:

1. Li, G.-W., Burkhardt, D., Gross, C., & Weissman, J. S. (2014). Quantifying absolute protein synthesis rates reveals principles underlying allocation of cellular resources. *Cell*, 157(3), 624–635. <http://doi.org/10.1016/j.cell.2014.02.033>
2. Vischer, N. O. E., Verheul, J., Postma, M., van den Berg van Saparoea, B., Galli, E., Natale, P., et al. (2015). Cell age dependent concentration of Escherichia coli divisome proteins analyzed with ImageJ and ObjectJ. *Frontiers in Microbiology*, 6, 586. <http://doi.org/10.3389/fmicb.2015.00586>

Given that immunofluorescence can be prone to artifacts resulting from lack of specificity of antibodies, localization data should be supported by using a fluorescent derivative of Nlpl, fused to a protein that folds in the periplasm, like mNeonGreen.

We fully agree with the referee and explored this option at length. We have expressed GFP-Nlpl, Nlpl-GFP and Nlpl-GGS-mNG (see table 1), none of these constructs were able to fully restore the morphology of a Δnpl mutant to that of a wild-type strain (R3b, R3c and table 1). However, we gain similar results when complementing with a non-tagged Nlpl using a pBAD30 overexpression system (Fig. S12).

More importantly, we observed that fluorescently tagged Nlpl localizes as inclusion bodies even when its expression is not induced and independent of the growth conditions (temperature, media) we tried (Fig. R3a and R4 and table 1). We investigated this further to test if fluorescently tagged Nlpl is prone to proteolytic degradation. We harvested cells

expressing fluorescently tagged versions of Nlpl and performed a Western Blot using anti-Nlpl (Fig. R5). Our data suggests that Nlpl forms inclusion bodies that are partly degraded. This could be because of the multitude of interactions Nlpl has with other proteins being affected by the fluorescent tag. In conclusion, we hope that these findings will convince the reviewer that we cannot use fluorescently tagged Nlpl.

Figure R3. Nlpl-mNG does not fully complement the morphology of a Δnpl strain.

The BW25113 Δnpl strain was transformed with a plasmid expressing Nlpl-GGS-mNG from a weakened p_{trc99A} promoter (3). The cells were grown in minimal glucose (Gb4) medium at 28°C and induced for 2 mass doublings with IPTG as indicated. Cells were imaged live or fixed. The localisation of Nlpl-GGS-mNG expressed from a medium copy number plasmid from a weakened p_{trc99A} promoter without induction after fixation is reminiscent of the localization pattern revealed by immunolabeling of cells (Fig. A). The scale bar equals 2 μ m. However, the construct only partly complements the morphology of the Δnpl phenotype (Fig. B and C), each point represents an average of \sim 1000 cells. This partly complementation is probably due to breakdown products of the fusions (see Fig R4 immunoblot).

Sample	Diameter	SD	Number of repeats
Empty plasmid	0.72	0	2
Nlpl-NG	0.8	0.0058	3
Nlpl-NG 5 μ M IPTG	0.83	0.03	3
BW25113	0.86		1

Table 1. FP fusions to NlpI do not complement.

strain	plasmid	length	sd	dia	stdev	n
BW ΔnlpI	none	4.79	1.14	0.98	0.05	944
BW25113	none	4.94	1.14	1.12	0.05	564
MG1655	none	5.04	1.13	1.06	0.05	828
MG1655 $\Delta nlpI$	pGFP-NlpI	4.26	0.98	1	0.05	393
MG1655 $\Delta nlpI$	pGFP-NlpI IPTG	4.03	1.01	0.99	0.06	755
MG1655 $\Delta nlpI$	NlpI-GFP IPTG	5.22	1.49	0.97	0.05	135
MG1655:: nlpI-gfp	none	5.58	1.73	0.89	0.04	185

Figure R4. NlpI-mNG does not complement the morphology of a $\Delta nlpI$ strain.

The MG1655 $\Delta nlpI$ strain was transformed with plasmids (*pDSW210*) expressing NlpI-GFP and GFP-NlpI from an IPTG inducible promoter. The cells were grown in TY medium at 37°C and induced for 20 min 50 μ M IPTG. Cells were imaged live. NlpI-GFP localizes as inclusion bodies and GFP-NlpI localizes in the periplasm and as inclusion bodies. The scale bar equals 5 μ m. However, the constructs only partly complement the morphology of the $\Delta nlpI$ phenotype (Table 1). This partly complementation is probably due to breakdown products of the fusions (see Fig R4 immunoblot).

Figure R5. Fluorescently tagged NlpI is proteolytically degraded.

Immunoblot of strains grown in TY at 37°C. The cells were harvested by centrifugation and immediately boiled in sample buffer and applied on an SDS-PAGE gel of 10%. Immunoblotting with anti-NlpI and development by ECL shows that all fusions are degraded, which might explain their partial complementation of the $\Delta nlpI$ phenotype. The various forms of NlpI are indicated by *.

References:

3. Blaauwen, den, T., Aarsman, M. E. G., Vischer, N. O. E., & Nanninga, N. (2003). Penicillin-binding protein PBP2 of *Escherichia coli* localizes preferentially in the lateral wall and at mid-cell in comparison with the old cell pole. *Molecular Microbiology*, 47(2), 539–547

This is particularly important because the authors claim that NlpI interacts with divisome proteins (Line 158), but no enrichment of NlpI at the septum is observed (line 262), which seems contradictory.

Although we provide *in vitro* evidence that NlpI interacts with many division proteins, the reviewer is correct that we do not provide evidence where these interactions occur in the cell and if the nature of these interactions is direct, stable or transient. Therefore, we rephrased the discussion to address this point.

2. Why was the effect of NlpI on the activity of EPases assessed using different substrates (sacculi or pre-digested mucopeptides) for different samples? Fig 3a should clearly indicate which substrate was used in which experiment, given that all data is presented in a single graph.

The effect of NlpI on the activity of EPases was tested against substrate which each respective EPase was previously found to have activity against. The details about which substrate was used for each experiment is described in materials and methods, and we now mention the substrates in the figure legend according to the referee's suggestion.

3. In line 251-252 authors say that the presence of more MepS decreases cell width. They indeed show that lack of MepS results in thicker cells, but that does not necessarily imply the reverse. Have authors tested the effect of an overexpression of MepS? Also, given that MepS was suggested to have a role in cell elongation, it would be useful to show measurements for cell length.

We agree with the referee that the potential role of MepS in modulating cell morphology has not been explored in great detail in this manuscript or any other published work. We therefore teamed up with the group of Sven van Teeffelen (Institut Pasteur, Paris) who constructed a $\Delta mepS$ strain harboring a MepS encoding-plasmid (pBAD30) that is under the control of an arabinose inducible promoter. In brief, when MepS expression is repressed, cells are wider, similarly as in the mutation. Overexpression of MepS leads to a slight decrease in cell diameter, albeit lower than that seen in the $\Delta nlpI$ strain, although MepS levels are very high. We take this as additional evidence that the NlpI role goes beyond MepS. We added this new data in Fig. 3 and S13 and adjusted the text in the results and discussion accordingly, and added the new authors to the manuscript.

4. Was strain fitness calculated based on "colony integral opacity" (line 855) or "colony size" (Line 1251)?

We thank the reviewer for spotting this mistake: the strain fitness is calculated based on colony integral opacity in all cases and this has been corrected in the manuscript.

5. It is surprising that fitness and morphology defects do not correlate (fig 3 b and c), as the MepS mutant and NlpI/MepS double mutant are not severely impaired in fitness but have very thick cells. Do authors have an explanation for this?

Changes in cell shape can arise from different problems. Some of them are coupled with severe growth defects, especially when the envelope integrity is compromised. Others are not. For example, point mutations on MreB can change the width of *E. coli* from 10% thinner to almost double the wild-type width; these changes have almost no effect on batch/single-cell growth rate in rich media (see Shi *et al.*, Current Biol 2017 – PMID 29103935).

In Fig. 3b we observe some small fitness defects for $\Delta mepS$ (0.92) and $\Delta mepS\Delta nlpI$ (0.87) over wild-type growth. However, both strains seem to be doing quite well in terms of cell envelope integrity (CPRG assay – Fig. 3h). In contrast, the *nlpI* mutant combines severe morphology defects, with compromised envelope, and hence the fitness effects are stronger.

Minor concerns

Line 180 - It would help readers that may be less familiar with the technique, to very briefly explain the principle of MST in the context of the aim of the experiments performed.

A brief explanation of the principal behind MST has been added (line 181-184). This section now explicitly explains that the use of MST (titrating a fluorescently labelled protein against a serial dilution of unlabelled ligand) was to assay for binding of the two proteins and to obtain apparent binding affinities for these interactions.

Line 847-850 - It is not very clear how the strains are plated. Is the initial rectangular plate, inoculated with 200ul of culture (spread with glass beads) used for the Rotor HAD replicator to get the inoculum for the genetic interaction (final) plates? If so, this could be explained a bit more clearly.

Yes. A sentence to clarify the use of the source plate for the genetic interaction assay has been added.

Line 1206 - typo in word "hydrolases"

This has been corrected

Line 1211 - state meaning of FDR in legend

The meaning of FDR (False discovery rate) is now stated in the legend.

Fig S5d - show complete western image, to enable evaluation of the specificity of antibodies.

We show now the complete row of the Western Blot image.

Referee #2:

Banzhaf et al. examined the function of lipoprotein Nlpl in E. coli as a scaffold and regulatory protein for peptidoglycan degradation and synthesis enzymes. 2D-TT and affinity chromatography approaches suggested interactions of Nlpl with MepS, LdtF, PBP4, and EnvC. Mutations of nlpl or nlpl together with various endopeptidases result in changes in cell morphology that remain unexplained. Nlpl was added to various endopeptidases and tested for the ability to cleave peptide-crosslinked tetra-tetra peptidoglycan dimers. Nlpl inhibited endopeptidase MepM but had little or no effect on other endopeptidases in vitro. Double mutants lacking nlpl and a biosynthetic PBP or PBP activator showed substantial growth

defects. Overall, these studies increase our understanding of protein-protein interactions in *E. coli* that relate to peptidoglycan metabolism. However, the results showing Nlpl decreasing activities of two endopeptidases are inconsistent with the model of Nlpl functioning to bring endopeptidases to a cell wall site to allow PG synthesis. Thus the reader is left with an unclear idea of Nlpl function.

1. Statistical tests for significance should be applied to the data in Fig. 2c, 2d, and 3a and that information should be supplied in the figure.

The data were fitted using a linear model. Calculated means were compared using Tukey's HSD test, resulting in p-values corrected for multiple testing. Relevant p-values are highlighted directly in the figure, all p-values can be found in the supplementary table 7 and 8.

2. A major question in these studies is what the function of Nlpl is for PG degradation and synthesis. Clearly, Nlpl results in less MepS and thus less endopeptidase activity by that enzyme, and binding of MepM by Nlpl caused less dimer degradation (Fig 3a). Thus, Nlpl may function to decrease endopeptidase activities. However, the authors propose that Nlpl is functioning to bring endopeptidases to the site of PG synthesis so that they can separate PG strands (lines 453-456 and elsewhere). The data show that PG synthetic enzymes associate with Nlpl, but questions remain about the outcome of these interactions. Do the trimeric complexes involving Nlpl show increased degradation of PG dimers? Would mutants where the synthetic enzymes can't interact with the Nlpl complex show less PG synthesis, more or less endopeptidase activity, or something else? It is simply unclear at the end of these studies what the effect of Nlpl is on PG synthesis and degradation.

This is a valid point. In response to this question we tested the possibility that a complex containing MepS-Nlpl and synthetic enzymes (PBP1A/LpoA), would result in an effect on synthesis activity. As MepS may potentially degrade products of the PG synthesis assay, we performed the assays using a catalytically inactive MepS^{C68A} (MepS*).

First, we tested whether presence of MepS* and Nlpl have any effects on the glycosyltransferase (GTase) activity of PBP1A ± LpoA using two different GTase assays (Figure S9a-d) and found in both cases there was no significant effect. Second, we carried out an HPLC-based activity assay using radiolabelled lipid-II substrate to test for effects on GTase and transpeptidase (TPase) activity and similarly observed no major effects of MepS*-Nlpl on this (Fig. S9e, Fig. S10). Overall, we could not detect differences of adding Nlpl/MepS to *in vitro* PG assays. It is possible that we are missing some component that would result into changes of the enzymes tested, or that our end-point TPase assay cannot

capture differences in enzymatic rates. However, as discussed in the manuscript, we think that Nlpl's main function is to scaffold PG EPases (and maybe other PG hydrolases).

In addition, we tested if PG composition of Δnpl mutants is altered compared to PG isolated of wild type cells. In this analysis, we did not find changes in the overall PG composition (Fig. S11), which may not be surprising. As many PG enzymes have overlapping functions in PG metabolism often changes in morphology do not correlate with changes in PG composition.

In summary, our data does not evidence for an direct, active role of Nlpl on PG synthesis. Nevertheless, these findings are consistent with our proposed model that Nlpl acts as a scaffolding protein, where it facilitates the formation of PG-biosynthetic complexes. In a context where synthetic enzymes cannot interact with Nlpl, we would hence speculate that the respective activities won't necessarily be affected, but that the lack of coordination of synthesis and hydrolysis activities might result in problems for the cell. Our conclusion for the current paper is hence that Nlpl has roles in coordinating PG enzymes or acting as an adaptor for PG hydrolases within PG multi-protein complexes; but that further mechanistic work (which is beyond the scope of the current paper) will be needed to explore the outstanding questions (i.e., whether Nlpl regulates/coordinates other groups of hydrolases, whether Nlpl becomes a stronger activator/repressor of one type of hydrolytic enzymes in the presence of others, and what the exact effects on PG biosynthesis are). Trying to obtain answers to these questions will form the basis of subsequent studies and publications and we hope that the referees and editors will agree that these remaining questions do not detract from the quality of work done in the present study.

3. The most interesting biological results from these studies are the morphological changes and growth defects in the *npl* and endopeptidase mutants, but the reasons for these defects were not determined.

We agree with the referee that the morphological changes of Δnpl and Δ endopeptidase mutants are intriguing. We rewrote this section and added evidence that some of the morphological changes in the *npl* mutant are partially due to elevated MepS levels (Fig. 3d-f), but cannot be solely explained by that. The morphological profiling revealed some further milder interactions of *npl* with *mepM* (Fig. 3c and S12c), which were left undiscussed in the first version and are now discussed in the text (and are consistent with Nlpl interacting with and the repressing the activity of MepM). In terms of fitness and envelope integrity the Δnpl problems are resolved when knocking out MepS in conditions we tested (new Fig. 3g-f), and we hence attribute them to higher MepS levels.

At the moment we don't know the exact source for the additional morphological defect (MepS independent) of $\Delta nlpI\Delta mepS$ or $\Delta nlpI$ cells (which are thinner than cells overexpressing MepS). However, we take this as evidence that the NlpI role goes beyond MepS, which is corroborated by TPP, pull-down, interaction and genetic interactions with PG synthases. We hope that these observations will be the starting point of further studies that go beyond the scope of this manuscript and have different directions.

4. I think the idea that this report is the first example of PG synthases and hydrolases in a multi-enzyme complex with a regulatory lipoprotein (lines 310-312) is incorrect as the PBP3-AmiC-NlpD complex has been described (starting with Heidrich C et al. 2001 Mol Microbiol).

We thank the referee for this remark. It is indeed correct that there has previously been other biochemical reconstitution of trimeric complexes and hence we have removed the sentence saying we have resolved the first complex containing PG synthases, hydrolases and lipoproteins; and instead replaced it with a reference to the actual first work (Vollmer *et al.* 1999 J Biol Chem) and instead state that our work provides further evidence for the existence of complexes containing PG synthases, hydrolases and lipoproteins.

5. More information should be provided on the known interactions of *E. coli* endopeptidases with other PG breakdown or synthesis proteins, such as those from Höltje's studies (Romeis and Höltje JBC (1994) is one example).

This is a good suggestion by the referee and it also provides more general context to where research in the field is currently at, prior to our work. We have therefore added few sentences and references to cite previous work which show that besides interactions with NlpI, the respective *E. coli* EPases have interactions with other PG processing proteins which could also contribute to the coordination of their activity.

6. LpoA and LpoB are specific to gamma-proteobacteria, and nlpI is not found in all Gram-negative species. The authors should make it clear how widespread and conserved NlpI is so that the reader will know how generally this model can be applied.

We thank the reviewer for this suggestion. We tried to do such a study prior to the first submission but were unable to get a clear answer to this. NlpI domain analysis reveals three TPR (Tetratricopeptide repeat) motifs that could be used to elucidate this. However, TPR motifs/domains are very common throughout the phylogenetic tree of life (used to mediate protein-protein interactions), and almost all organisms carry several such proteins. Therefore, defining NlpI conservation by sequence or structural similarity gets difficult and can produce misleading results. For example, the initial publication that led to the structure

determination of Nlpl found close structural similarity to human proteins and an *E. coli* maltose transcriptional regulator (Wilson C *et al.*, FEBS J 2004). Both matches were due to the TPR-like content of these proteins.

Minor points.

1. The authors should not be iffy in the discussion of their results. Lines 321, 343, and 361 all contain words that make it sound like the authors don't trust their data and don't think that the proteins actually bind one another. They measured dissociation constants, so they know the proteins bind to each other.

The words that suggest uncertainty in these sections have been removed to express a greater degree of confidence in our conclusions.

2. The amidase regulators and their effects on their specific interacting partners have been described, so the "appear to have specificity" phrase in lines 387-388, is an inaccurate description of what is known to occur (Uehara T *et al.* 2010. EMBO J).

The sentence now reads "Nlpl is more promiscuous than the previously identified regulators EnvC and NlpD, which bind to their cognate amidases" to reflect more accurately the known mechanism of amidase activation by EnvC and NlpD

Referee #3

This ambitious, excellent study by Drs. Banzhaf *et al.* presents new, intriguing data about a fundamental question in peptidoglycan (PG) cell wall synthesis in gram-negative bacteria, using *E. coli* as a model. The paper focuses on the function of the outer membrane NplI protein, which previous studies had shown plays a role in targeting the MepS PG endopeptidase hydrolase for degradation by a periplasmic protease. The data in this new paper greatly extends the findings from these previous studies by showing that NplI functions in setting the cellular amount and stability of numerous other proteins, including key proteins involved in PG synthesis and hydrolytic remodeling. Numerous key findings are succinctly summarized by the headings of each section of the Results. These results are based on an extensive study using biochemical, physiological, and microscopic methods. Together, these experiments provide some of the first direct evidence in support of the Holtje model that PG synthases and hydrolases function together in a physical complex during PG synthesis. This paper strongly suggests that NplI serves as a key adaptor in coordinating complex formation between PG synthases and hydrolases in these complexes, although many details about this process remain to be addressed in future studies.

The data content in this paper is truly impressive and is based on several biochemical approaches, including high-level TMT proteomics and biochemical demonstration of tripartite

complexes containing NplI and other proteins along with activity assays. These data indicate that NplI has two general functions: setting the stability of partner proteins and affecting enzymatic activity, such as inhibition of the MepM endopeptidase. Straightforward genetic experiments also indicate a nearly synthetic lethal relationship between mutations in *nplI* and *mreB/lpoB*, suggesting a requirement of NplI for function of the remaining PBP1A/LpoA synthase proteins. The large amount of data is of high quality and uniformly convincing from multiple biological replicates with appropriate statistics.

This is a well-presented, important paper that will influence the field about this central process of coordination of PG synthesis machines. There are a couple points of interpretation and presentation that if addressed, would add to this paper.

Major points

1. Lines 1, lines 102, and 459 and perhaps elsewhere. "Nucleation" has a specific biochemical meaning in the context in polymer formation that does not really apply here. Rather, NplI acts a key player in "partner switching" regulation among various PG synthesis and hydrolase proteins. Therefore, please consider changing nucleates/nucleation to "partnering" or "adapting" (e.g., in lines 459: "by way of partnering proteins in complexes containing synthases...")

We changed "nucleation" to "adaptor", "coordinator" or "scaffolding", depending on context throughout the whole manuscript.

2. Line 82. Does the combined data set indicate why filamentation occurs in $\Delta nplI$ mutants at higher temperatures or low osmolarity? This point might be mentioned in the Discussion.

We do not provide strong evidence that can explain why $\Delta nplI$ mutants filament under those conditions. However, the interaction of Nlpl with several division proteins suggest that Nlpl could affect cell division, beyond its capacity to regulate MepS levels that in this study we linked to impact cell width. As suggested, we added this aspect to the discussion.

3. Line 84. Please consider changing "hypervesiculation" to "increased membrane vesical formation".

We changed the term according to the reviewer's suggestion.

4. Section starting on line 106 and Figure 1. The paper focuses on the effects of $\Delta nplI$ mutants on the abundance and stability of proteins involved in PG biosynthesis. However, there are many very light gray dots in Figure 1 based on the supplemental tables. That is,

the delta-npII mutation is much more pleiotropic than stressed in the paper. It is understandable to focus on PG synthesis, but this broad pleiotropy should be mentioned in the Results and commented upon in the Discussion. Can this broadly pleiotropic effect be related to other studies where PG synthesis or OM assembly was disrupted by other means?

We fully agree with the comment and reworded this section to stronger highlight the pleiotropic effects of $\Delta nlpI$ mutants. We are not aware of other studies that used 2D-TPP proteomics or similar methods able to monitor abundance and stability *in vivo* that would be a good reference point to benchmark our results.

5. Related to point 4, it should be commented on whether the npII mutation can be cleanly complemented by ectopic expression of npII+ and that the pleiotropy of the delta-npII mutation is not due to polarity.

We addressed this and added this information into the section. In summary, we cannot fully restore the morphology of an $\Delta nlpI$ mutant by expressing endogenous NlpI from an arabinose inducible, low copy number plasmid (pBAD30), but complementation is almost complete (Fig. S12 a and b). Therefore, we cannot fully exclude polarity effects, but other reasons such as deregulated NlpI or MepS levels and/or problems with protein folding/translocation when NlpI is mildly overexpressed could be reason for this slight difference.

6. Line 123-124). Some of the abundance changes detected by proteomics are small/moderate (e.g., PBP1A, MltG, etc.). Were these smaller changes in protein amounts corroborated by quantitative western blotting?

We agree with the reviewer that these changes were relatively small, albeit reproducible (quantified in replicates by generally more than 5 peptides per protein). We have therefore modified the text accordingly. Since the manuscript focuses on NlpI and its capability to act as an adaptor of endopeptidases, we feel that further validation of these changes is beyond the scope of this manuscript.

7. Lines 156-166. Why was MepS (and MepM; line 184) not detected in the pull-down experiments? This apparently inconsistent observation is mentioned later in the Discussion, but should be mentioned here and commented upon. Is it because MepS amount is really low in the npII+ bacteria due to degradation? Was the pull down shown in Figures 1C and 1D performed for the delta-npII mutant to get at proteins whose amounts might be greatly decreased due to targeted degradation mediated by binding to NpII?

MepS and MepM could not be detected in the applied fraction of the affinity chromatography by MS. As suggested by the reviewer this might be due to degradation or general low cellular levels of MepS and MepM. Their small size creates an additional problem for MS. Please note that both are almost undetectable in wildtype cells (TPP), and MepS gets at much higher levels when *nlpI* is knocked out.

We added the sentence in the result section

“For both affinity chromatography experiments we were unable to detect the known NlpI binding partner MepS in the applied extract, likely due to low cellular levels of MepS“

8. Related to point 4, there seem to be a lot of other proteins (more gray dots) that are pulled down by binding to NpII in Figures 1C and 1D. It seems that NpII binds to a lot of proteins. Can anything else be said about the implications of these data? Are there any other patterns that emerge?

We thank the referee for this remark. It is true that NlpI pulls down many proteins which could be indirect or unspecific binders. Although, we have obtained similar results with other bait proteins in the past, NlpI may be particularly increased in unspecific/indirect bindings due to the 3 TPR domains it carries. Nevertheless, to emphasize why we chose to highlight only known peptidoglycan proteins, we now show the enrichment analysis on the pulled down proteins, which point to this direction. We added those in the main text (see below) and all data can be seen in Supplementary tables 9 and 10.

NlpI pulled down proteins that are significantly enriched in the GO terms, “cell cycle”, “cell wall organisation”, “cell division”, “peptidoglycan metabolic processes” and other relevant envelope biogenesis GO terms.

9. Line 284-285. To a lesser extent, NpII complexes are also important to the function of PBP1B/LpoB as well. This point might be stated. Also, in line 277, was the nearly synthetic lethal relationship between delta-npII and delta-mreB/lpoB detected in the previous lacZ-based screen by Paradis-Bleau et al? Please comment on.

Paradis-Bleau *et al.* did not disclose the full hit-list of their screen. They state, “We screened a total of 30,000 colonies and isolated 16 slb mutants. Less than half of these (5/16) were completely dependent on IPTG for growth. Three of the IPTG-dependent mutants had transposon insertions that mapped to the gene for PBP1a (*ponA*), indicating that the screen worked as expected (Fig. 2D). Two of the remaining IPTG-dependent mutants had

transposons that mapped within lpoA (yraM) (Fig. 2C and 2D)". The authors did not further comment on the 11 hits that are not fully IPTG independent (*i.e.*, synthetic sick).

10. Figure 6 and Discussion. Some of the major patterns might be pulled together even more in summary Figure 6. As noted in above, the data suggest that NpII can play two roles: affecting stability and/or abundance or changing enzymatic activity, which was assayed directly for a couple of endopeptidases. These roles might be added to Fig. 6a as two columns. Changes in relative stability and/or abundance in the absence of NpII of each protein listed can be taken from data in Figure 1a and 1b. This compilation would make it really clear that for some EPases, NpII plays a major destabilizing role (*i.e.*, MepS), but not others (*i.e.*, PBP4, Fig. 1b and line 217). For the endopeptidases that were assayed, another column could indicate whether direct NpII binding affected activity or did not.

We added the requested changes in Figure 6a and included the data of Figure 1a and 1b in the summary. However, we felt that adding information of the activity assays does not improve the figure to justify the additional space needed.

11. Methods. Based on the assays, it seems that the biochemical assays containing purified NpII were performed with solubilized NpII in the absence of outer membrane binding. This point should be qualified in the Results.

We indeed used soluble NpI for all the MST, AUC and activity assays and have now added a sentence saying "A soluble version of NpI lacking its membrane anchor was used for all these assays" (lines 176-177).

Minor points

12, Line 228. Consider changing: "on its own" to "by itself".

This has been changed.

13. Line 256. Omit: "probably".

This has been changed.

14. Line 262. Consider changing to: "with certain divisome proteins".

This has been changed.

15. Line 364. Change to: "facilitates the proteolytic".

This has been changed.

16. Line 388. Consider adding a paragraph break after: "2009)."

This has been changed.

17. Line 390. Omit: "also".

This has been omitted.

18. Line 408. Omit: "them"

This has been omitted.

19. Line 450. Change to: ", because the latter is".

This has been changed.

20. Section starting with line 520. Mention the antibody specificity controls done for the immunofluorescence microscopy.

This has been addressed.

21. Line 786. Comma missing before "500".

This has been rectified.

22. Tables 1, 2, and 3 could be moved to the Supplemental Materials.

Tables were moved into supplementary information.

Thank you for submitting a revised version of your manuscript. It has now been seen by all original referees, who appreciate the breadth of the revision and now are broadly in favour of publication of the study. However, I would like to ask you to address a few mainly editorial issues before I can extend formal acceptance of the manuscript:

1. Please address the remaining minor comments from reviewer #3.
2. Please check the attached manuscript text file with minor edits in the figure legend section from our data editor Georgi Hristov that require your approval and addition of some statistical details.
3. Please submit up to five keywords.
4. Please remove the Significance section or integrate the text in the Abstract or Introduction sections.
5. Please update the nomenclature of Appendix Figures and Tables to that of Appendix Figure S1/Appendix Table S1 etc.
6. Please update the reference format to up to 20 authors before et al (<https://www.embopress.org/page/journal/14602075/authorguide#referencesformat>)
7. Please make sure that the proteomics data generated in the study are deposited at a public database before online publication of the manuscript. Please also reformat the Data Availability section in Materials and Methods according to our guidelines (<https://www.embopress.org/page/journal/14602075/authorguide#datadeposition>).
8. Figure 2B appears to be re-used in Appendix Figure S2B. The same is the case for Appendix Figures S6B and S8A. Please indicate in the figure that the data stem from the same experiment.
9. For the Supplementary Tables you would like to include in the final manuscript (I presume that some tables will be removed upon data deposition in a database), please rename the tables uploaded as datasets into Table EV1 etc and add a short title and table legend to the respective file in a separate tab. Some tables appear to represent source data - please indicate this so that we can assign the files accordingly.
10. In Figure S7A, the middle and the bottom row profiles fully overlap and appear duplicated. I am not familiar with this type of data - is this coincidental?
11. In Figure 3B and 4B, the same plate overview image is used, while the magnifications show different colony constellations. I presume that the plate overview is shown mainly for illustrative purposes, therefore please add a note to that effect in the figure legends to avoid any possible post-publication misunderstandings.
12. During our routine text plagiarism analysis, I noticed that there are a few sentences in the Materials and Methods section that have been taken almost verbatim from another publication. I have attached the screenshots here. Please add a reference to the original article in these sections.
13. Papers published in The EMBO Journal include a 'Synopsis' to further enhance discoverability. Synopses are displayed on the html version of the paper and are freely accessible to all readers. The synopsis includes a short introductory paragraph, as well as 2-5 one-sentence bullet points that summarise the paper. Please send us your suggestions for the synopsis text and a synopsis image (size max 550x400 pixels).

You can use the link below to upload the revised manuscript. Please let me know if you have any further questions regarding any of these points.

Thank you again for giving us the chance to consider your manuscript for The EMBO Journal. I am

Referee #1:

The authors have done an enormous amount of extra work to try to answer all reviewers comments and should be complimented for that. However, unfortunately, most of the experiments did not result in increasing clarification of the in vivo role of Nlpl. Specifically, Figs S9, S10 and S11, show no difference of PBP1A-LopA activity, PG synthesis or composition in the presence or in the absence of Nlpl. Efforts to localize Nlpl fused to mNeonGreen in live cells were not successful. Figures R1 and R2 (included in the point by point review) show no difference in the localization of PBP1A and PBP1B in the presence or absence of Nlpl. In this later case however, this may be due to the low quality of the images. One can barely see any specific localization of PBP1A and PBP1B even in the presence of Nlpl, while there is published data from the senior author where this localization is much more clear. Other authors have also shown enrichment of PBP1B at the septum (besides the periphery) using GFP fusions (eg doi: 10.7554/eLife.40754). If the PBP1A/B localization is not evident in the presence of Nlpl, it is not surprising that no change is observed. These experiments could be repeated using strains that express fluorescent derivatives of only one of the PBPs.

One alternative to these localization studies would be to determine the effect of Nlpl absence on the dynamics of PBP1A, by single molecule imaging. The group of KC Huang has shown that "Perturbations to PBP1A activity, either directly through antibiotics or indirectly through PBP1A's interaction with its lipoprotein activator or other synthases, shifted the fraction of mobile molecules" (DOI <https://doi.org/10.1038/ncomms13170>). Something similar could be expected in the absence of Nlpl.

However, I recognize that these are not trivial experiments and that the cellular role of Nlpl can be further explored in a subsequent paper.

Therefore, despite the fact that I find it intriguing that a protein with a central role in coordinating PG synthases and hydrolases has no effect on various of the phenotypes tested, and that I agree with reviewer #2 on the fact that the function of Nlpl in the cell is still not clear, I consider the in vitro data solid enough for publication.

Referee #2:

The revised manuscript presents a more thorough and more nuanced description of the roles of Nlpl in scaffolding and possibly regulating endopeptidases and other peptidoglycan degradation enzymes in *E. coli*. The work is impressive in scope and detail. These studies further advance the understanding of the processes necessary for building and modifying the sacculus.

Referee #3:

The authors did a rigorous and thoughtful job of addressing each of the previous review points with changes and additional experiments. This is an outstanding experimental story about Npll phenotypes and interactions. The paper is notably data rich and complete in its descriptions of the wide range of sophisticated approaches used. The conclusions reached are reasonable and based on careful interpretations of the data at this point. The paper generates several ideas and hypotheses about interactions and mechanisms that can be followed up for specific interactors in future studies.

There are a couple of minor typos to fix. Also, there are a couple of points in the discussion that might be elaborated on a bit more.. These changes can be made rapidly by simple and limited changes to this well-written, important paper.

1. Line 103. Please insert "(TPR)" after "repeats", since this is the first time this abbreviation is defined .
2. Line 244. Please consider changing "stimulated or repressed" to "increased or decreased", because "repressed" usually has a genetic connotation.
3. Line 272. Please consider changing "but" to "and".
3. Line 273. Please consider changing "problems" to some other word, such as "phenotypes".
4. Line 285. Please consider rewording: "... 3D), but the cell morphological changes only to a certain extent."
5. L. 299 and 300. Please remove one "therefore".
6. Line 485-486, lines 501-504, and lines 512-515. This section becomes a bit too fixated on Npll as "the" adaptor for PG synthesis. Figure 1 seems to indicate that there may be several other proteins involved in this process, along with Npll. Did any other adapter candidates emerge from the

ontology analysis? The likelihood of other regulatory/adaptor protein components could be brought out a bit more directly.

7. Lines 512- 521. The lack of concentration of NplI at septa seems to be argued around too much, as if it is not quite right. The pull down experiments in high salt indicate complexes in non-synchronized cells, but do not necessarily show direct interactions. Given the genetic links to the Class A PBPs and their regulators and their distribution over the cell bodies, why couldn't NplI be involved mainly in elongosome synthesis or timing elongation versus septal synthesis? This point might be touched on a bit more.

8. Line 484. Do double mutants of delta-nlpl and some of the endopeptidase mutants change PG composition more than the single mutants? That is, could the lack of primary effects of delta-nlpl on PG composition reflect redundancy of the endopeptidases and/or Class A PBPs? Please consider commenting on this point, because it is a bit unexpected that there is no PG change in the delta-nlpl single mutant.

Editorial points

1. Please address the remaining minor comments from reviewer #3.

Response: We modified the manuscript according to the suggestions made by reviewer 3. Below we provide a point-by-point response to these points.

2. Please check the attached manuscript text file with minor edits in the figure legend section from our data editor Georgi Hristov that require your approval and addition of some statistical details.

Response: We checked the manuscript file, approved the edits and added statistical data. We also adjusted the legends to Appendix figures accordingly.

3. Please submit up to five keywords.

Response: Bacterial cell envelope; Peptidoglycan; Peptidoglycan hydrolases; Outer membrane lipoprotein NlpI; *Escherichia coli*.

4. Please remove the Significance section or integrate the text in the Abstract or Introduction sections.

Response: Significance was removed.

5. Please update the nomenclature of Appendix Figures and Tables to that of Appendix Figure S1/Appendix Table S1 etc.

Response: We renamed the Appendix Figures and Tables and uploaded the others tables as data sets according to point 9 below.

6. Please update the reference format to up to 20 authors before et al

(<https://www.embopress.org/page/journal/14602075/authorguide#referencesformat>)

Response: We updated the references.

7. Please make sure that the proteomics data generated in the study are deposited at a public database before online publication of the manuscript. Please also reformat the Data Availability section in Materials and Methods according to our guidelines

(<https://www.embopress.org/page/journal/14602075/authorguide#datadeposition>).

Response: We added the following information to Materials and Methods:

Section Mass spectrometry to identify NlpI affinity chromatography hits:

"The mass spectrometry proteomics data have been deposited to the ProteomeXchange Consortium via the PRIDE partner repository with the dataset identifier PXD016825."

Section 2D-TT data analysis:

"The mass spectrometry proteomics data have been deposited to the ProteomeXchange Consortium via the PRIDE partner repository with the dataset identifier PXD016819."

8. Figure 2B appears to be re-used in Appendix Figure S2B. The same is the case for Appendix Figures S6B and S8A. Please indicate in the figure that the data stem from the same experiment.

Response: This is correct and we now indicate this in the figure legends:

Legend to Fig. 2B. "B. MepS-NlpI interaction by MST as an example plot for Fig 2A. The same plot is also shown in Fig S2B. Fl, fluorescently labelled; FNorm, normalized fluorescence.

Legend to Fig. S8A. "A. MST interaction curves of PBP4 with PBP1A, LpoA and NlpI, respectively. The PBP4-LpoA interaction plot was already shown in Fig. S6B and is displayed here again for completeness."

9. For the Supplementary Tables you would like to include in the final manuscript (I presume that some tables will be removed upon data deposition in a database), please rename the tables uploaded

as datasets into Table EV1 etc and add a short title and table legend to the respective file in a separate tab. Some tables appear to represent source data - please indicate this so that we can assign the files accordingly.

Response: We kept all tables because we the raw MS data uploaded to PRIDE are before we define thresholds and call hits. Our tables show the normalized and quantified MS data. To clarify this we renamed the MS data shown in the manuscript into “complete data after quantification”.

We renamed all data tables, also in the main text. They now have descriptive names and we also display this information within the Excel file.

10. In Figure S7A, the middle and the bottom row profiles fully overlap and appear duplicated. I am not familiar with this type of data - is this coincidental?

Response: The editor was right, many thanks for catching this mistake (indeed, we added one of the scans twice). We corrected the figure.

11. In Figure 3B and 4B, the same plate overview image is used, while the magnifications show different colony constellations. I presume that the plate overview is shown mainly for illustrative purposes, therefore please add a note to that effect in the figure legends to avoid any possible post-publication misunderstandings.

Response: The plates are examples and we now indicate this in the legend to Fig. 3B and 4B: “An example of a 384 plate is shown.”

12. During our routine text plagiarism analysis, I noticed that there are a few sentences in the Materials and Methods section that have been taken almost verbatim from another publication. I have attached the screenshots here. Please add a reference to the original article in these sections.

Response: We are extremely thankful for pointing to these mistakes and are extremely sorry for not paying proper attention to this. We now cite the original paper for these methods.

13. Papers published in The EMBO Journal include a 'Synopsis' to further enhance discoverability. Synopses are displayed on the html version of the paper and are freely accessible to all readers. The synopsis includes a short introductory paragraph, as well as 2-5 one-sentence bullet points that summarise the paper. Please send us your suggestions for the synopsis text and a synopsis image (size max 550x400 pixels).

Response: We added a synopsis.

Comments by the reviewers

Response: We would like to thank all reviewers for their help in improving our manuscript and their positive comments about the revised version. We thank reviewer 1 for suggesting to measure the dynamics of single molecules of PBP1A. This and other experiments on the cellular role of NlpI will be part of follow-up studies.

Specific points made by reviewer 3

1. Line 103. Please insert "(TPR)" after "repeats", since this is the first time this abbreviation is defined .

Response: done

2. Line 244. Please consider changing "stimulated or repressed" to "increased or decreased", because "repressed" usually has a genetic connotation.

Response: changed.

3. Line 272. Please consider changing "but" to "and".

Response: changed.

3. Line 273. Please consider changing "problems" to some other word, such as "phenotypes".

Response: changed.

4. Line 285. Please consider rewording: "... 3D), but the cell morphological changes only to a certain extent."

Response: changed.

5. L. 299 and 300. Please remove one "therefore".

Response: removed.

6. Line 485-486, lines 501-504, and lines 512-515. This section becomes a bit too fixated on NlpI as "the" adaptor for PG synthesis. Figure 1 seems to indicate that there may be several other proteins involved in this process, along with NlpI. Did any other adapter candidates emerge from the ontology analysis? The likelihood of other regulatory/adaptor protein components could be brought out a bit more directly.

Response: The reviewer is correct with this point and we did not meant to write that NlpI is the only adaptor protein for PG synthesis. To clarify this we changed the sentence to "... NlpI or other adaptor proteins,..." (line 508).

7. Lines 512- 521. The lack of concentration of NlpI at septa seems to be argued around too much, as if it is not quite right. The pull down experiments in high salt indicate complexes in non-synchronized cells, but do not necessarily show direct interactions. Given the genetic links to the Class A PBPs and their regulators and their distribution over the cell bodies, why couldn't NlpI be involved mainly in elongosome synthesis or timing elongation versus septal synthesis? This point might be touched on a bit more.

Response: We clarified the part to say that NlpI could have a main role in cell elongation.

8. Line 484. Do double mutants of delta-nlpI and some of the endopeptidase mutants change PG composition more than the single mutants? That is, could the lack of primary effects of delta-nlpI on PG composition reflect redundancy of the endopeptidases and/or Class A PBPs? Please consider commenting on this point, because it is a bit unexpected that there is no PG change in the delta-nlpI single mutant.

Response: We did not analyse the PG composition of multiple mutants. The reviewer is right that the lack of an effect of the nlpI deletion on PG composition could be due to redundancy in PG synthases and hydrolases. At this time we would like to refrain from commenting on this further as the comment would be too speculative.

Thank you for implementing the final revisions in your manuscript. I apologise for the delay in communicating the decision due to the post-holiday backlog. The main issues have now been addressed and I am pleased to inform you that your manuscript has been accepted for publication.

Corresponding Author Name: Waldemar Vollmer

Journal submitted to: EMBO Journal

Manuscript number:EMBOJ-2019-102246R